# Exploiting Local Convergence of Quasi-Newton Methods Globally: Adaptive Sample Size Approach

**Qiujiang Jin**
ECE Department
The University of Texas at Austin
Austin, TX 78712, USA
qiujiang@austin.utexas.edu

**Aryan Mokhtari**
ECE Department
The University of Texas at Austin
Austin, TX 78712, USA
mokhtari@austin.utexas.edu

## Abstract

In this paper, we study the application of quasi-Newton methods for solving empirical risk minimization (ERM) problems defined over a large dataset. Traditional deterministic and stochastic quasi-Newton methods can be executed to solve such problems; however, it is known that their global convergence rate may not be better than first-order methods, and their local superlinear convergence only appears towards the end of the learning process. In this paper, we use an adaptive sample size scheme that exploits the superlinear convergence of quasi-Newton methods globally and throughout the entire learning process. The main idea of the proposed adaptive sample size algorithms is to start with a small subset of data points and solve their corresponding ERM problem within its statistical accuracy, and then enlarge the sample size geometrically and use the optimal solution of the problem corresponding to the smaller set as an initial point for solving the subsequent ERM problem with more samples. We show that if the initial sample size is sufficiently large and we use quasi-Newton methods to solve each subproblem, the subproblems can be solved superlinearly fast (after at most three iterations), as we guarantee that the iterates always stay within a neighborhood that quasi-Newton methods converge superlinearly. Numerical experiments on various datasets confirm our theoretical results and demonstrate the computational advantages of our method.

## 1 Introduction

In various machine learning problems, we aim to find an optimal model that minimizes the expected loss with respect to the probability distribution that generates data points, which is often referred to as expected risk minimization. In realistic settings, the underlying probability distribution of data is unknown and we only have access to a finite number of samples ($N$ samples) that are drawn independently according to the data distribution. As a result, we often settle for minimizing the sample average loss, which is also known as empirical risk minimization (ERM). The gap between expected risk and empirical risk which is known as generalization error is deeply studied in the statistical learning literature [1–5], and it is well-known that the gap between these two loss functions vanishes as the number of samples $N$ in the ERM problem becomes larger.

Several variance reduced first-order methods have been developed to efficiently solve the ERM problem associated with a large dataset [6–13]. However, a common shortcoming of these *first-order* methods is that their performance heavily depends on the problem condition number $\kappa := L/\mu$, where $L$ is the risk function smoothness parameter and $\mu$ is the risk function strong convexity constant. As a result, these first-order methods suffer from slow convergence when the problem has a large condition number, which is often the case in datasets with thousands of features. In fact, the best-known bound

Table 1: Comparison of adaptive sample size methods, in terms of number of gradient computations, Hessian computations, matrix-vector product evaluations and matrix inversions.

| Algorithm | gradient comp. | Hessian comp. | matrix-vec product | matrix inversion |
|---|---|---|---|---|
| Ada Newton | $\mathcal{O}(N)$ | $\mathcal{O}(N)$ | $\mathcal{O}(\log N)$ | $\mathcal{O}(\log N)$ |
| AdaQN | $\mathcal{O}(N)$ | $m_0$ | $\mathcal{O}(\log N)$ | 1 |

achieved by first-order methods belongs to Katyusha [11] which has a cost of $\mathcal{O}((N + \sqrt{\kappa N}) \log N)$ to achieve the statistical accuracy of an ERM problem with $N$ samples.

To resolve this issue, second-order methods such as Newton's method and quasi-Newton methods are often used, as they improve curvature approximation by exploiting second-order information. Several efficient implementations of second-order methods for solving large-scale ERM problems have been proposed recently, including incremental Newton-type methods [14], sub-sampled Newton algorithms [15], and stochastic quasi-Newton methods [16–20]. However, these methods still face two major issues. First, their global convergence requires selection of a small step size which slows down the convergence, or implementation of a line-search scheme which is computationally prohibitive for large-scale ERM problems, since it requires multiple passes over data. The second disadvantage of these (deterministic and stochastic) second-order methods is that their local quadratic or superlinear convergence only appears in a small local neighborhood of the optimal solution. This often means that, when we solve an ERM problem, the fast superlinear or quadratic convergence rate of these methods happens *when the iterates have already reached the statistical accuracy of the problem* and the test accuracy would not improve, even if the ERM problem error (training error) decreases.

To address these two shortcomings of second-order methods for solving large-scale ERM problems, the adaptive sample size Newton (Ada Newton) method was proposed in [21]. In adaptive sample size methods, instead of solving the ERM corresponding to the full training set directly (as in deterministic methods) or using a subset of samples at each iteration (as in stochastic methods), we minimize a sequence of geometrically increasing ERM problems. Specifically, we solve the ERM problem corresponding to a subset of samples, e.g. $m$ samples, within their statistical accuracy, then we use the resulting solution as an initial point for the next ERM problem with $n = \alpha m$ samples ($\alpha > 1$), where the larger set with $n$ samples contains the smaller set with $m$ samples. The main principle of this idea is that all training samples are drawn from the same distribution, therefore the optimal values of the ERM problems with $m$ and $n$ samples are close to each other. The main result of Ada Newton in [21] is that if one chooses the expansion factor $\alpha$ properly, the solution for the problem with $m$ samples is within the Newton quadratic convergence neighborhood of the next ERM problem with $n = \alpha m$ samples. Hence, each subproblem can be solved quickly by exploiting the quadratic convergence of Newton's method at every step of the learning process. This approach is promising as the overall number of gradient and Hessian computations for Ada Newton is $\mathcal{O}(N)$, while it requires computing $\mathcal{O}(\log N)$ Newton's directions. Note that $\mathcal{O}$ notation only hides absolute constants and these complexities are independent of problem condition number $\kappa$. The main drawback of this approach, however, is the requirement of $\mathcal{O}(\log N)$ Newton's directions computation, where each could cost $\mathcal{O}(d^3)$ arithmetic operations, where $d$ is the dimension. It also requires $\mathcal{O}(N)$ Hessian evaluations which requires $\mathcal{O}(Nd^2)$ arithmetic operations, and this could be computationally costly.

A natural approach to reduce the computational cost of Newton's method is the use of quasi-Newton algorithms, in which, instead of exact computation of the Hessian and its inverse required for Newton update, we approximate the objective function Hessian and its inverse only by computing first-order information (gradients). Since this approximation only requires matrix-vector product computations at each iteration, the computational complexity of quasi-Newton methods is reduced. It is known that quasi-Newton methods such as DFP and BFGS converge superlinearly in a neighborhood close to the optimal solution, i.e., the iterates $\{\mathbf{w}_k\}$ converge to the optimal solution $\mathbf{w}^*$ at a rate of $\lim_{k \to +\infty} \frac{\|\mathbf{w}_{k+1} - \mathbf{w}^*\|}{\|\mathbf{w}_k - \mathbf{w}^*\|} = 0$, where $k$ is the iteration number. To exploit quasi-Newton methods for the described adaptive sample size scheme, explicit superlinear convergence rate of these methods is needed to characterize the number of iterations required for solving each ERM problem. The non-asymptotic convergence rate of quasi-Newton methods was unknown until very recently, when a set of concurrent papers [22, 23] and [24] showed that in a local neighborhood of the optimal solution, the iterates of DFP and BFGS converge to the optimal solution at a rate of $(1/k)^{k/2}$.

In this paper, we exploit the finite-time analysis of the quasi-Newton methods studied in [24] to develop an adaptive sample size quasi-Newton (AdaQN) method that reaches the statistical accuracy

of the ERM corresponding to the full training set after $\mathcal{O}(N)$ gradient computations and $\mathcal{O}(\log N)$ matrix-vector product evaluations and only 1 matrix inversion. In Table 1, we formally compare the computation cost of our proposed AdaQN algorithm and the Ada Newton method in [21]. As shown in Table 1, the total number of gradient evaluations and matrix-vector product computations for AdaQN are the same as the ones for Ada Newton, while the number of Hessian computations for AdaQN and Ada Newton are $m_0$ and $\mathcal{O}(N)$, respectively, where $m_0$ is the size of initial training set in the adaptive sample size scheme, and it is substantially smaller than $N$, i.e., $m_0 \ll N$. In our main theorem we show that $m_0$ is lower bounded by $\Omega\left(\max\left\{d, \kappa^2 s \log d\right\}\right)$, where $s$ is a parameter determined by the loss function and the structures of the training data (see details in Remark 2). Hence, in this paper, we focus on the regime that the total number of samples satisfies $N \gg \max\left\{d, \kappa^2 s \log d\right\}$. However, our numerical experiments in Section 5 show that AdaQN often achieves small risk for $m_0$ much smaller than this lower bound and the effectiveness of AdaQN goes beyond this regime. Also, the number of times that Ada Newton requires to compute the inverse of a square matrix (or solving a linear system) is $\mathcal{O}(\log N)$, while to implement AdaQN we only require one matrix inversion. We should also add that the implementation of Ada Newton requires a backtracking mechanism for the choice of growth factor $\alpha$. This backtracking scheme is needed to ensure that the size of training set does not grow too fast, and the iterates stay within the quadratic convergence neighborhood Newton's method for next ERM problem. Unlike Ada Newton, our proposed AdaQN method does not require such backtracking scheme, and as long as the size of initial training set $m_0$ is sufficiently large, we can double the size of training set at the end of each phase, i.e., we set $\alpha = 2$.

It is worth noting that in [25], the authors study a similar approach to our proposed adaptive sample size quasi-Newton method called the batch-expansion training (BET) method. The BET method differs from our proposed framework in terms of algorithm development and convergence analysis. Specifically, the BET method uses a similar idea of geometrically increasing the size of the training set, while using the limited-memory BFGS (L-BFGS) method for solving the Empirical Risk Minimization (ERM) subproblems. The authors leverage the linear convergence rate of L-BFGS to characterize the overall complexity of the BET algorithm. We should mention that since the linear convergence rate of L-BFGS is not provably better than first-order methods, the overall iteration complexity of BET is similar to the one for adaptive sample size first-order methods [26] which depends on the problem condition number. In contrast to [25], in this paper, we leverage the local *superlinear* convergence rate of the BFGS method to improve the overall complexity of adaptive sample size first-order methods, and hence, it also improves the complexity of the BET method.

## 2   Problem formulation

In this paper, we focus on minimizing large-scale empirical risk minimization (ERM) problems. Consider the loss function $f : \mathbb{R}^d \times \mathcal{Z} \to \mathbb{R}$ that takes inputs as the decision variable $\mathbf{w} \in \mathbb{R}^d$ and random variable $\mathbf{Z} \in \mathcal{Z}$ with a probability distribution $P$. The expected risk minimization problem aims at minimizing the expected loss over all possible choices of $\mathbf{Z}$,

$$\mathbf{w}^* := \arg\min_{\mathbf{w}} R(\mathbf{w}) = \arg\min_{\mathbf{w}} \mathbb{E}_{\mathbf{Z} \sim P}[f(\mathbf{w}; \mathbf{Z})]. \tag{1}$$

Note that in this expression we defined the expected risk $R : \mathbb{R}^d \to \mathbb{R}$ as $R(\mathbf{w}) := \mathbb{E}_{\mathbf{Z} \sim P}[f(\mathbf{w}; \mathbf{Z})]$ and $\mathbf{w}^*$ as an optimal solution of the expected risk minimization. In this paper we focus on loss functions $f$ that are strongly convex with respect to $\mathbf{w}$, hence the optimal solution $\mathbf{w}^*$ is unique.

The probability distribution $P$ is often unknown, and we only have access to a finite number of realizations of the random variable $\mathbf{Z}$, which we refer to as our training set $\mathcal{T} = \{\mathbf{z}_1, \ldots, \mathbf{z}_N\}$. These $N$ realizations are assumed to be drawn independently and according to $P$. Hence, instead of minimizing the expected risk in (1), we attempt to minimize the empirical risk corresponding to the training set $\mathcal{T} = \{\mathbf{z}_1, \ldots, \mathbf{z}_N\}$. To formalize this, let us first define $\mathcal{S}_n = \{\mathbf{z}_1, \ldots, \mathbf{z}_n\}$ as a subset of the training set $\mathcal{T}$ which contains its first $n$ elements. Without loss of generality we defined an ordering for the elements of the training set $\mathcal{T}$. The empirical risk corresponding to the set $\mathcal{S}_n$ is defined as $R_n(\mathbf{w}) := \frac{1}{n} \sum_{i=1}^{n} f(\mathbf{w}; \mathbf{z}_i)$ and its optimal solution $\mathbf{w}_n^*$ is given by

$$\mathbf{w}_n^* := \arg\min_{\mathbf{w}} R_n(\mathbf{w}) = \arg\min_{\mathbf{w}} \frac{1}{n} \sum_{i=1}^{n} f(\mathbf{w}; \mathbf{z}_i). \tag{2}$$

Note that the ERM problem associated with the full training set $\mathcal{T}$ is a special case of (2) when $n = N$, and its unique optimal solution is denoted by $\mathbf{w}_N^*$.

The gap between the optimal values of empirical risk $R_n$ and expected risk $R$ is well-studied in the statistical learning literature and here we assume the following upper bound holds

$$\mathbb{E}\left[|R_n(\mathbf{w}_n^*) - R(\mathbf{w}^*)|\right] \leq V_n, \tag{3}$$

where $V_n$ is a function of the sample size $n$ that approaches zero as $n$ becomes large. The expectation in (3) is over the set $\mathcal{S}_n$. In this paper, we assume that all the expectations are with respect to the corresponding training set. Classic results have established bounds of the form $V_n = \mathcal{O}(1/\sqrt{n})$ [1, 27] and other recent papers including [2–5] show that under stronger regularity conditions such as strong convexity, we have $V_n = \mathcal{O}(1/n)$, for $n = \Omega(\kappa^2 \log d)$. In this paper, we assume the requirements for the bound $V_n = \mathcal{O}(1/n)$ are satisfied.

Since the gap between the optimal empirical and expected risks is always bounded above by the error $V_n$, there is no point to reducing the optimization error of minimizing $R_n$ beyond the statistical accuracy $V_n$. In other words, if we obtain a solution $\hat{\mathbf{w}}$ such that $\mathbb{E}[R_n(\hat{\mathbf{w}}) - R_n(\mathbf{w}_n^*)] = \mathcal{O}(V_n)$, there would be no benefit in further minimizing $R_n$. Due to this observation, we say $\hat{\mathbf{w}}$ solves the ERM in (2) within its *statistical accuracy* if $\mathbb{E}[R_n(\hat{\mathbf{w}}) - R_n(\mathbf{w}_n^*)] \leq V_n$. The ultimate goal is to efficiently find a solution $\mathbf{w}_N$ that reaches the statistical accuracy of the full training set $\mathcal{T}$, i.e., $\mathbb{E}[R_N(\mathbf{w}_N) - R_N(\mathbf{w}_N^*)] \leq V_N$. Next, we state the notations and assumptions.

**Assumption 1.** *For all values of $\mathbf{z}$, the loss function $f(\mathbf{w}; \mathbf{z})$ is twice differentiable and $\mu$-strongly convex with respect to $\mathbf{w}$ and its gradient is Lipschitz continuous with parameter $L > 0$.*

**Assumption 2.** *For all values of $\mathbf{z}$, the loss function $f(\mathbf{w}; \mathbf{z})$ is self-concordant with respect to $\mathbf{w}$.*

Assumptions 1 and 2 are customary for the analysis of quasi-Newton methods. These conditions also imply that the empirical risk $R_n$ is also self-concordant, strongly convex with $\mu$, and its gradient is Lipschitz continuous with $L$. Hence, the condition number of $R_n$ is $\kappa := L/\mu$.

**Computational cost notation**  We report the overall computational cost in terms of these parameters: (i) $\tau_{grad}$ and $\tau_{Hess}$ which denote the cost of computing one gradient of size $d$ and one Hessian of size $d \times d$; (ii) $\tau_{prod}$ which indicates the cost of computing the product of a square matrix of size $d \times d$ with a vector of size $d$; and (iii) $\tau_{inv}$ which denotes the cost of computing the inverse of a square matrix of size $d \times d$ or the cost of solving a linear system with $d$ variables and $d$ equations.

## 3  Algorithm

**Quasi-Newton methods**  Before introducing our method, we first briefly recap the update of quasi-Newton (QN) methods. Given the current iterate $\mathbf{w}$, the QN update for the ERM problem in (2) is

$$\mathbf{w}^+ = \mathbf{w} - \eta \mathbf{H}\, \nabla R_n(\mathbf{w}), \tag{4}$$

where $\eta > 0$ is a step size and $\mathbf{H} \in \mathbb{R}^{d \times d}$ is a symmetric positive definite matrix approximating the Hessian inverse $\nabla^2 R_n(\mathbf{w})^{-1}$. The main goal of quasi-Newton schemes is to ensure that the matrix $\mathbf{H}$ always stays close to $\nabla^2 R_n(\mathbf{w})^{-1}$. There are several different approaches for updating the Hessian approximation matrix $\mathbf{H}$, but the two most-widely used updates are the DFP method defined as

$$\mathbf{H}^+ = \mathbf{H} - \frac{\mathbf{H}\mathbf{y}\mathbf{y}^\top \mathbf{H}}{\mathbf{y}^\top \mathbf{H}\mathbf{y}} + \frac{\mathbf{s}\mathbf{s}^\top}{\mathbf{s}^\top \mathbf{y}}, \tag{5}$$

and the BFGS update defined as

$$\mathbf{H}^+ = \left(\mathbf{I} - \frac{\mathbf{s}\mathbf{y}^\top}{\mathbf{s}^\top \mathbf{y}}\right) \mathbf{H} \left(\mathbf{I} - \frac{\mathbf{y}\mathbf{s}^\top}{\mathbf{s}^\top \mathbf{y}}\right) + \frac{\mathbf{s}\mathbf{s}^\top}{\mathbf{s}^\top \mathbf{y}}, \tag{6}$$

where $\mathbf{s} := \mathbf{w}^+ - \mathbf{w}$ is the variable variation and $\mathbf{y} := \nabla R_n(\mathbf{w}^+) - \nabla R_n(\mathbf{w})$ is the gradient variation. If we follow the updates in (5) or (6), then finding the new Hessian inverse approximation and consequently the new descent direction $-\mathbf{H}^+ \nabla R_n(\mathbf{w}^+)$ only require computing a few matrix-vector multiplications. Considering this point and the fact that each step of BFGS or DFP requires $n$ gradients evaluations, the computational cost of each step of BFGS and DFP is $\mathcal{O}(n\tau_{grad} + \tau_{prod})$.

---

**Algorithm 1** AdaQN

---

**Input:** The initial sample size $m_0$; The initial argument $\mathbf{w}_{m_0}$ within the statistical accuracy of $R_{m_0}$; The initial Hessian inverse $\mathbf{H}_{m_0} = \nabla^2 R_{m_0}(\mathbf{w}_{m_0})^{-1}$;

1: Set $n \leftarrow m_0$;
2: **while** $n \leq N$ **do**
3:     Set the initial argument $\hat{\mathbf{w}} \leftarrow \mathbf{w}_n$, the initial matrix $\hat{\mathbf{H}} \leftarrow \mathbf{H}_{m_0}$ and $iteration = 1$;
4:     Increase the sample size: $n \leftarrow \min\{2n, N\}$;
5:     **while** $iteration \leq t_n$ **do**
6:         $\hat{\mathbf{w}}^+ \leftarrow \hat{\mathbf{w}} - \hat{\mathbf{H}} \nabla R_n(\hat{\mathbf{w}})$;
7:         $\mathbf{s} \leftarrow \hat{\mathbf{w}}^+ - \hat{\mathbf{w}}$;
8:         $\mathbf{y} \leftarrow \nabla R_n(\hat{\mathbf{w}}^+) - \nabla R_n(\hat{\mathbf{w}})$;
9:         $\hat{\mathbf{H}}^+ = \left(\mathbf{I} - \frac{\mathbf{s}\mathbf{y}^\top}{\mathbf{s}^\top \mathbf{y}}\right) \hat{\mathbf{H}} \left(\mathbf{I} - \frac{\mathbf{y}\mathbf{s}^\top}{\mathbf{s}^\top \mathbf{y}}\right) + \frac{\mathbf{s}\mathbf{s}^\top}{\mathbf{s}^\top \mathbf{y}}$;
10:         Set $\hat{\mathbf{w}} \leftarrow \hat{\mathbf{w}}^+$, $\hat{\mathbf{H}} \leftarrow \hat{\mathbf{H}}^+$ and $iteration \leftarrow iteration + 1$;
11:     **end while**
12:     Set $\mathbf{w}_n \leftarrow \hat{\mathbf{w}}$;
13: **end while**

---

### 3.1 Adaptive sample size quasi-Newton algorithm (AdaQN)

BFGS, DFP or other quasi-Newton (QN) methods can be used to solve the ERM problem corresponding to the training set $\mathcal{T}$ with $N$ samples, but (i) the cost per iteration would be of $\mathcal{O}(N\tau_{grad} + \tau_{prod})$, (ii) the superlinear convergence only appears towards the end of learning process when the iterates approach the optimal solution and statistical accuracy is already achieved, and (iii) they require a step size selection policy for their global convergence. To resolve the first issue and reduce the high computational cost of $\mathcal{O}(N\tau_{grad} + \tau_{prod})$, one could use stochastic or incremental QN methods that only use a subset of samples at each iteration; however, stochastic or incremental QN methods (similar to deterministic QN algorithms) outperform first-order methods only *when the iterates are close to the optimal solution* and they also require *line-search* schemes for selecting the step size. Hence, none of these schemes is able to exploit fast superlinear convergence rate of QN methods throughout the entire training process, while always using a constant step size of 1.

Next, we introduce the adaptive sample size quasi-Newton (AdaQN) method that addresses these drawbacks. Specifically, (i) AdaQN does not require any line-search scheme and (ii) exploits the superlinear rate of QN methods throughout the entire training process. In a nutshell, AdaQN leverages the interplay between the statistical accuracy of ERM problems and superlinear convergence neighborhood of QN methods to solve ERM problems efficiently. It uses the fact that all samples are drawn from the same distribution, therefore the solution for ERM with $m$ samples is not far from the solution of ERM with $n$ samples, where $n$ samples contain the $m$ samples. Hence, the solution of the problem with less samples can be used as a warm-start for the problem with more samples. Note that there are two important points here that we should highlight. First, if $m$ and $n$ are chosen properly, the solution of the ERM problem $R_m$ corresponding to the set $\mathcal{S}_m$ with $m$ samples will be in the superlinear convergence neighborhood of the next ERM problem corresponding to the set $\mathcal{S}_n$ with $n$ samples, where $\mathcal{S}_m \subset \mathcal{S}_n$. Hence, each subproblem can be solved with a few iterations of quasi-Newton methods (see Figure 5 in the appendix). Second, in each subproblem we only use a subset of samples, therefore the cost of running QN methods for solving subproblems with $m$ samples (where $m \ll N$) is significantly less than the update of QN methods for the full training set.

The steps of AdaQN are outlined in Algorithm 1. We start with a small subset of the full training set with $m_0$ samples and solve its corresponding ERM problem within its statistical accuracy. The initial ERM problem can be solved using any iterative method and its cost will be negligible as it scales with $m_0$ instead of $N$, where $m_0 \ll N$. In the main loop (Step 2-Step 12) we implement AdaQN. Specifically, we first use the solution from the previous round (or $\mathbf{w}_{m_0}$ when $n = m_0$) as the initial iterate, while we set the initial Hessian inverse approximation as $\hat{\mathbf{H}} = \mathbf{H}_{m_0} = \nabla^2 R_{m_0}(\mathbf{w}_{m_0})^{-1}$ (Step 3). Then, we double the size of the training set by adding more samples to the active training set (Step 4). In Steps 5-10 we run the BFGS update for minimizing the loss $R_n$, while we keep updating the iterates and Hessian inverse approximation. Once, the required condition for convergence specified in Step 5 is obtained, we output the iterate $\mathbf{w}_n$ as the iterate that minimizes $R_n$ within its

statistical accuracy. In Step 8 we used the update for BFGS, but one can simply replace this step with the update of the DFP method. As we ensure that iterates always stay within the neighborhood that BFGS (or DFP) converges superlinearly, the step size of these methods can be set as $\eta = 1$. We repeat this procedure until we reach the whole dataset $\mathcal{T}$ with $N$ samples. In Algorithm 1, for the phase that we minimize $R_n$ (Steps 5-10), our goal is to find a solution $\mathbf{w}_n$ that satisfies $\mathbb{E}\left[R_n(\mathbf{w}_n) - R_n(\mathbf{w}_n^*)\right] \leq V_n$. We use $t_n$ to indicate the maximum number of QN updates to find such a solution. Our theoretical result (Theorem 1) suggests that due to fast convergence of QN methods, at most $t_n = 3$ iterations required to solve each subproblem within its statistical accuracy.

**Remark 1.** *Note that a natural choice for the initial Hessian inverse approximation at each phase of AdaQN with $n$ samples is the Hessian inverse at the initial iterate of that phase, i.e., setting the initial Hessian inverse approximation as $\hat{\mathbf{H}} = \nabla^2 R_n(\mathbf{w}_m)^{-1}$, where $\mathbf{w}_m$ is the solution of the previous phase with $m = n/2$ samples. However, if we follow this initialization, then for each phase of AdaQN we need to compute $n$ new Hessians to evaluate $\nabla^2 R_n(\mathbf{w}_m)$ and one matrix inversion to find $\nabla^2 R_n(\mathbf{w}_m)^{-1}$, which would increase the computational cost of AdaQN. To avoid this issue, for all values of $n$, we always use the initial Hessian inverse approximation matrix corresponding to the first phase with $m_0$ samples and set $\hat{\mathbf{H}} = \mathbf{H}_{m_0} = \nabla^2 R_{m_0}(\mathbf{w}_{m_0})^{-1}$ (Step 3). We show that even under this initialization the required condition for superlinear convergence of DFP or BFGS is always satisfied if the initial sample size $m_0$ is large enough. Note that by following this scheme, we only need to compute $m_0$ Hessians and a single matrix inversion to implement AdaQN. This is a significant gain compared to Ada Newton in [21], which requires computing $\mathcal{O}(N)$ Hessians and $\mathcal{O}(\log N)$ matrix inversions.*

# 4 Convergence analysis

In this section, we characterize the overall complexity of AdaQN. We only state the results for BFGS defined in (6), as the proof techniques and overall complexity bounds for adaptive sample size versions of DFP and BFGS are similar.

First we state an upper bound for the sub-optimality of the variable $\mathbf{w}_m$ with respect to the empirical risk of $R_n$, given that $\mathbf{w}_m$ has achieved the statistical accuracy of the previous empirical risk $R_m$.

**Proposition 1.** *Consider $S_m$ and $S_n$ such that $S_m \subset S_n \subset \mathcal{T}$, where there are $m$ samples in $S_m$ and $n$ samples in $S_n$ and $n \geq m$. Consider the corresponding empirical risk functions $R_m$ and $R_n$ defined based on $S_m$ and $S_n$, respectively. Assume that $\mathbf{w}_m$ solves the ERM problem of $R_m$ within its statistical accuracy, i.e., $\mathbb{E}\left[R_m(\mathbf{w}_m) - R_m(\mathbf{w}_m^*)\right] \leq V_m$. Then we have*

$$\mathbb{E}\left[R_n(\mathbf{w}_m) - R_n(\mathbf{w}_n^*)\right] \leq 3V_m. \tag{7}$$

Proposition 1 characterizes the sub-optimality of the variable $\mathbf{w}_m$ for the empirical risk $R_n$ which plays a fundamental role in our analysis. We use this bound later to show $\mathbf{w}_m$ is close enough to $\mathbf{w}_n^*$ so that $\mathbf{w}_m$ is in the superlinear convergence neighborhood of $R_n$.

Next, we require a bound on the number of iterations needed by BFGS to solve a subproblem, when the initial iterate is within the superlinear convergence neighborhood. We establish this bound by leveraging the result of [24] which provides a non-asymptotic superlinear convergence for BFGS.

**Proposition 2.** *Consider AdaQN in the phase that the active training set contains $n$ samples. If Assumptions 1-2 hold and the initial iterate $\mathbf{w}_m$ and Hessian approximation $\nabla^2 R_{m_0}(\mathbf{w}_{m_0})$ satisfy*

$$\|\nabla^2 R_n(\mathbf{w}_n^*)^{\frac{1}{2}}(\mathbf{w}_m - \mathbf{w}_n^*)\| \leq \frac{1}{300},$$
$$\|\nabla^2 R_n(\mathbf{w}_n^*)^{-\frac{1}{2}}[\nabla^2 R_{m_0}(\mathbf{w}_{m_0}) - \nabla^2 R_n(\mathbf{w}_n^*)]\nabla^2 R_n(\mathbf{w}_n^*)^{-\frac{1}{2}}\|_F \leq \frac{1}{7}, \tag{8}$$

*then after $t_n$ iterations we achieve the output $\mathbf{w}_n$ with the following convergence result*

$$R_n(\mathbf{w}_n) - R_n(\mathbf{w}_n^*) \leq 1.1\left(\frac{1}{t_n}\right)^{t_n}[R_n(\mathbf{w}_m) - R_n(\mathbf{w}_n^*)]. \tag{9}$$

Proposition 2 shows that if the initial Hessian approximation error is small and the initial iterate is close to the optimal solution, then the iterates of BFGS with step size $\eta = 1$ converge to the optimal

solution at a superlinear rate of $(1/k)^k$ after $k$ iterations. The inequalities in (8) identify the required conditions to ensure that $\mathbf{w}_m$ is within the superlinear convergence neighborhood of BFGS for $R_n$.

In the next two propositions we show that if the initial sample size $m_0$ is sufficiently large, both $\mathbf{w}_m$ and $\nabla^2 R_{m_0}(\mathbf{w}_{m_0})$ satisfy the conditions in (8) in expectation which are required to achieve the local superlinear convergence.

**Proposition 3.** *Consider Algorithm 1 for the phase that the active training set contains $n$ samples with empirical risk $R_n$. Suppose Assumptions 1-2 hold, and further suppose we are given $\mathbf{w}_m$ which is within the statistical accuracy of $R_m$, i.e, $\mathbb{E}\left[R_m(\mathbf{w}_m) - R_m(\mathbf{w}_m^*)\right] \le V_m$, and $n = 2m$. If the sample size $m$ is lower bounded by $m = \Omega\left(\kappa^2 \log d\right)$, then*

$$\mathbb{E}\left[\|\nabla^2 R_n(\mathbf{w}_n^*)^{\frac{1}{2}}(\mathbf{w}_m - \mathbf{w}_n^*)\|\right] \le \frac{1}{300}. \tag{10}$$

Proposition 3 shows that if the training set size is sufficiently large, then the solution of the previous phase is within the BFGS superlinear convergence neighborhood of the current problem, and the first condition in (8) holds in expectation. This condition is indeed satisfied throughout the entire learning process, if it holds for the first training set with $m_0$ samples, i.e., $m_0 = \Omega\left(\kappa^2 \log d\right)$.

Next we establish under what condition the Hessian approximation used in adaptive sample size method which is the Hessian evaluated with respect to $m_0$ samples, i.e., $\nabla^2 R_{m_0}(\mathbf{w}_{m_0})$, satisfies the second condition in (8) in expectation which is required for the superlinear convergence of BFGS.

**Proposition 4.** *Consider AdaQN in the phase that the active training set contains $n$ samples. If Assumptions 1-2 hold and the initial sample size $m_0$ satisfies $m_0 = \Omega\left(\max\left\{d, \kappa^2 s \log d\right\}\right)$, where $s$ is defined as $s := \sup_{\mathbf{w},n}\left(\frac{\mathbb{E}\left[\|\nabla^2 R_n(\mathbf{w}) - \nabla^2 R(\mathbf{w})\|_F\right]}{\mathbb{E}\left[\|\nabla^2 R_n(\mathbf{w}) - \nabla^2 R(\mathbf{w})\|\right]}\right)^2$. Then for any $n \ge 2m_0$ we have*

$$\mathbb{E}\left[\|\nabla^2 R_n(\mathbf{w}_n^*)^{-\frac{1}{2}}[\nabla^2 R_{m_0}(\mathbf{w}_{m_0}) - \nabla^2 R_n(\mathbf{w}_n^*)]\nabla^2 R_n(\mathbf{w}_n^*)^{-\frac{1}{2}}\|_F\right] \le \frac{1}{7}. \tag{11}$$

Based on Proposition 4, if the size of initial training set satisfies $m_0 = \Omega\left(\max\left\{d, \kappa^2 s \log d\right\}\right)$, then the second condition in (8) holds in expectation. By combining the results of Propositions 3 and 4, we obtain the required conditions for $m_0$. Moreover, Proposition 2 quantifies the maximum number of iterations required to solve each ERM problem to its corresponding statistical accuracy. In the following theorem, we exploit these results to characterize the overall computational complexity of AdaQN to reach the statistical accuracy of the full training set with $N$ samples.

**Theorem 1.** *Consider AdaQN described in Algorithm 1 for the case that we use BFGS updates in (6). Suppose Assumptions 1-2 hold, and the initial sample size $m_0$ is lower bounded by*

$$m_0 = \Omega\left(\max\left\{d, \kappa^2 s \log d\right\}\right), \tag{12}$$

*where $s = \sup_{\mathbf{w},n}\left(\frac{\mathbb{E}\left[\|\nabla^2 R_n(\mathbf{w}) - \nabla^2 R(\mathbf{w})\|_F\right]}{\mathbb{E}\left[\|\nabla^2 R_n(\mathbf{w}) - \nabla^2 R(\mathbf{w})\|\right]}\right)^2$. Then at the stage with $n$ samples, AdaQN finds $\mathbf{w}_n$ within the statistical accuracy of $R_n$ after at most $\boxed{t_n = 3}$ iterations. Further, the computational cost of AdaQN to reach the statistical accuracy of the full training set $\mathcal{T}$ is*

$$\tau_{inv} + m_0 \tau_{Hess} + 6N \tau_{grad} + 3\left(1 + \log(N/m_0)\right)\tau_{prod}. \tag{13}$$

Theorem 1 states that if the initial sample size is sufficiently large, the number of required BFGS updates for solving each subproblem is at most 3 iterations. Further, it characterizes the overall computational cost of AdaQN.

**Remark 2.** *Since $\|\cdot\| \le \|\cdot\|_F \le \sqrt{d}\|\cdot\|$, parameter $s$ in Theorem 1 belongs to the interval $[1, d]$. Hence, in the worst case $s = d$ and $m_0 = \Omega\left(\kappa^2 d \log d\right)$. However, for many common classes of problems including linear regression and logistic regression and for training datasets with specific structures $s$ could be $\mathcal{O}(1)$. In those cases the initial sample size is lower bounded by $\Omega\left(\max\left\{d, \kappa^2 \log d\right\}\right)$; see Section C in the appendix for details. In fact, our numerical experiments also verify this observation and show that the choice of $m_0$ could be much smaller than the worst-case bound of $m_0 = \Omega\left(\kappa^2 d \log d\right)$; see Section 5.*

**Remark 3.** *In the above complexity bound, we neglect the cost of finding the initial solution $\mathbf{w}_{m_0}$, as the cost of finding an approximate solution for the first ERM problem with $m_0$ samples is negligible compared to the overall cost of the AdaQN. For instance, if one solves the first ERM problem using the Katyusha algorithm [11], the cost of the initial stage would be $\mathcal{O}\left(\tau_{grad}(m_0 + \sqrt{m_0\kappa_{m_0}})\log m_0\right)$ (where $\kappa_{m_0}$ is the condition number for ERM with $m_0$ samples), which is indeed dominated by the term $\mathcal{O}\left(N\tau_{grad}\right)$, when we are in the regime that $N \gg m_0$.*

**Remark 4.** *If one uses Katyusha, which is an optimal first-order method, to solve an ERM with $N$ samples, the overall gradient computation to achieve accuracy $\epsilon = V_N$ would be $\mathcal{O}((N + \sqrt{N\kappa})\log N)$. The gradient computation cost of AdaQN considering the initialization step is $\mathcal{O}(((m_0 + \sqrt{m_0\kappa})\log m_0) + N)$. Indeed, if we are in the regime that the size of initial set $m_0$ is sufficiently smaller than the size of full training set $N$, (i.e., $N \gg m_0 \log m_0$), then AdaQN gradient complexity scales as $\mathcal{O}(N + \sqrt{m_0\kappa}\log m_0)$ which is smaller than $\mathcal{O}(N\log N + \sqrt{N\kappa}\log N)$ cost of Katyusha. However, AdaQN requires evaluating one additional matrix inversion, $m_0$ Hessian evaluations and $\mathcal{O}(\log(N/m_0))$ matrix-vector product computations.*

## 5 Numerical experiments

Next, we practically evaluate the performance of AdaQN to solve large-scale ERM problems. We consider a binary classification problem with $l_2$ regularized logistic regression loss function, where $\mu > 0$ is the regularization parameter. The logistic loss is convex, and, therefore, all functions in our experiments are $\mu$-strongly convex. We normalize all data points with the unit norm so that the loss function gradient is Lipschitz continuous with $L \leq \mu + 1$, hence condition number is $\kappa \leq 1 + 1/\mu$. Moreover, the logistic regression loss function is self-concordant. Thus, Assumptions 1-2 hold.

We compare AdaQN with adaptive sample size Newton method (Ada Newton) [21], standard BFGS quasi-Newton method, the L-BFGS quasi-Newton method [28], the stochastic quasi-Newton (SQN) method proposed in [29], and three stochastic first-order methods, including stochastic gradient descent (SGD), the Katyusha algorithm [11], and SAGA which is a variance reduced method [7]. In our experiments, we start with the initial point $\mathbf{w}_0 = c * \vec{1}$ where $c > 0$ is a tuned parameter and $\vec{1} \in \mathbb{R}^d$ is the one vector. First we conduct several iterations of the gradient descent method on the initial sub-problem with $m_0$ samples until the initial condition $\|\nabla R_{m_0}(\mathbf{w}_{m_0})\| \leq \sqrt{2\mu V_{m_0}}$ is satisfied. Note that $R_{m_0}(\mathbf{w}_{m_0}) - R_{m_0}(\mathbf{w}_{m_0}^*) \leq \frac{1}{2\mu}\|\nabla R_{m_0}(\mathbf{w}_{m_0})\|^2$ implies that this initial condition guarantees $\mathbf{w}_{m_0}$ is within the statistical accuracy of $R_{m_0}$. We use $\mathbf{w}_{m_0}$ as the initial point of AdaQN and Ada Newton as presented in Algorithm 1. We use $\mathbf{w}_0 = c * \vec{1}$ as the initial point of all other algorithms. We include the cost of finding the proper initialization for AdaQN in our comparisons. We compare these methods over (i) MNIST dataset of handwritten digits [30], (ii) Epsilon dataset from PASCAL challenge 2008 [31], (iii) GISETTE handwritten digit classification dataset from the NIPS 2003 feature selection challenge [32] and (iv) Orange dataset of customer relationship management from KDD Cup 2009 [33].[1] More details provided in Table 2.

In our experiments, we observe that even when the initial sample size $m_0$ is smaller than the threshold in (12), AdaQN performs well and converges superlinearly in each subproblem. This is because (12) is a sufficient condition to guarantee our theoretical superlinear rate, and in practice smaller choices of $m_0$ also work. For Ada Newton we use the same scheme descried in Algorithm 1 and replace the QN update with Newton's method with step size 1. The step sizes of the standard BFGS method and the L-BFGS method are determined by the Wolfe condition [35] using the backtracking line search algorithm to guarantee they converge on the whole dataset. All hyper-parameters (initialization parameter $c$, step size, batch size, etc.) of BFGS, L-BFGS, stochastic quasi-Newton method, SGD, SAGA, and Katyusha have been tuned to achieve the best performance on each dataset.

The convergence results are shown in Figures 1-4 for the considered datasets. We report both training error, i.e., $R_N(\mathbf{w}) - R_N(\mathbf{w}_N^*)$, and test error for all algorithms in terms of number of effective passes over dataset and in terms of runtime. In general AdaQN mostly outperforms the first-order optimization methods (SGD, Katyusha and SAGA). This is caused by the fact that our considered problems are highly ill-conditioned. For instance, for the results of Epsilon dataset which has $\kappa \approx 10^4$, there is a substantial gap between the performance of AdaQN and first-order methods.

---

[1] We use LIBSVM [34] with license: `https://www.csie.ntu.edu.tw/~cjlin/libsvm/COPYRIGHT`.

Table 2: Datasets information: sample size $N$, dimension $d$, initial set size $m_0$ and regularization $\mu$.

| Dataset | $N$ | $d$ | $m_0$ | $\mu$ |
|---------|-----|-----|-------|-------|
| MNIST | 11774 | 784 | 1024 | 0.05 |
| GISETTE | 6000 | 5000 | 1024 | 0.05 |
| Orange | 40000 | 14000 | 8192 | 0.1 |
| Epsilon | 80000 | 2000 | 4096 | 0.0001 |

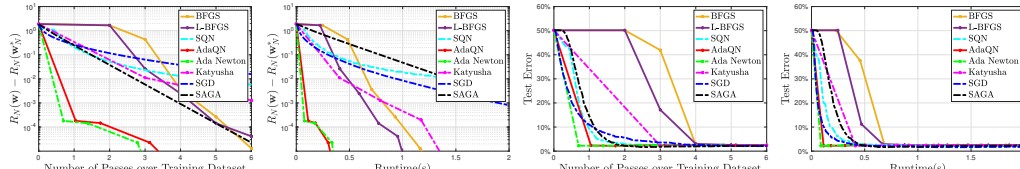

Figure 1: Training error (first two plots) and test error (last two plots) in terms of number of passes over dataset and runtime for MNIST dataset.

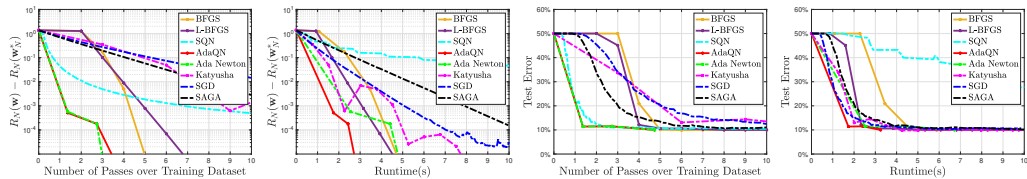

Figure 2: Training error (first two plots) and test error (last two plots) in terms of number of passes over dataset and runtime for GISETTE dataset.

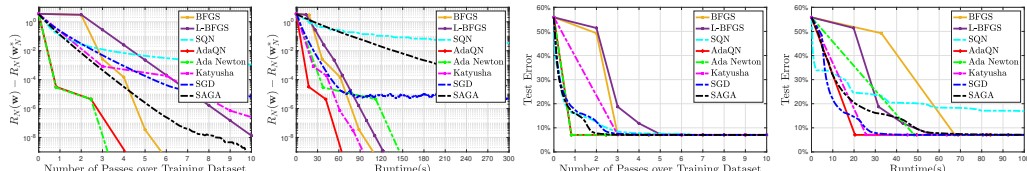

Figure 3: Training error (first two plots) and test error (last two plots) in terms of number of passes over dataset and runtime for Orange dataset.

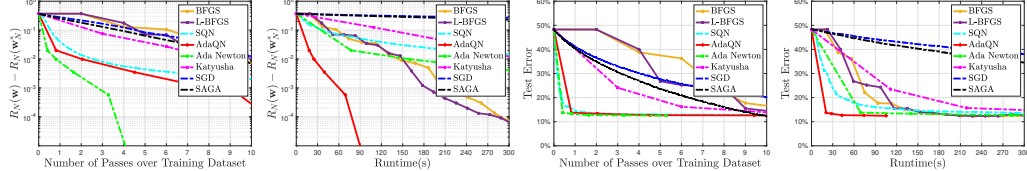

Figure 4: Training error (first two plots) and test error (last two plots) in terms of number of passes over dataset and runtime for Epsilon dataset.

We observe that AdaQN outperforms BFGS, L-BFGS and SQN in all considered settings. As discussed earlier, this observation is expected since AdaQN is the only quasi-Newton algorithm among these methods that exploits superlinear convergence of quasi-Newton methods throughout the entire training process. Moreover, AdaQN does not require a line search scheme, while to obtain the best performance of BFGS and L-BFGS, we used a line-search scheme which is often computationally costly.

Next we compare Ada Newton and AdaQN. As shown in Figure 1, for the MNIST dataset with low dimension ($d = 784$) and moderate number of samples ($N \approx 11000$), AdaQN performs very similar to Ada Newton. However, for settings where either the problem dimension $d$ or the number of samples $N$ is very large AdaQN outperforms Ada Newton significantly in terms of runtime. For instance, in Figures 2 and 3, which correspond to GISETTE dataset with $N = 6000$ samples and $d = 5000$ features and Orange dataset with $N = 40000$ samples and $d = 14000$ features, Ada Newton and AdaQN have almost similar convergence paths when compared in terms of number of passes over data, but in terms of runtime AdaQN is substantially faster than Ada Newton. This gap

comes from the fact that in these two examples the problem dimension $d$ is large and as a result the cost of inverting a matrix at each iteration, which is required for Ada Newton, is prohibitive.

For Epsilon dataset that has a relatively large number of samples $N = 80,000$ and moderate dimension $d = 2000$, as shown in Figure 4, Ada Newton outperforms AdaQN in terms of number of passes over data in both training and test errors. However, their relative performance is reversed when we compare them in terms of runtime. This time the slow runtime convergence of Ada Newton is due to the fact that sample size $N$ is very large. Note that Ada Newton requires $2N$ Hessian computations, while for AdaQN we only need $m_0$ Hessian evaluations.

**Remark 5.** *Theorem 1 requires the size of the initial training set $m_0$ to satisfy the condition in* (12). *In our experiments, however, we observe that $m_0$ could be much smaller than the threshold in* (12). *We believe that the main reason for this mismatch is that the recent non-asymptotic convergence analysis of BFGS that we exploit in our analysis may not be tight, and there is room for improving this bound. Specifically, establishing a less strict condition on the initial Hessian approximation error of BFGS required for achieving a superlinear convergence rate could lead to a much smaller lower bound for the parameter. Another possible reason for the gap between our theory and experiments is that our analysis is a worst-case analysis and it studies a lower bound that works for any setting. There could be some hard instances or corner cases that our current experiments do not capture, and for these hard instances we might need the number of samples required by our theoretical study. Identifying hard instances of empirical risk minimization problems for which a larger initial sample size is required is a research direction that requires further investigation.*

## 6   Discussion and future work

We proposed AdaQN algorithm that leverages the interplay between the superlinear convergence of quasi-Newton methods and statistical accuracy of ERM problems to efficiently minimize the ERM corresponding to a large dataset. We showed that if the initial sample size $m_0$ is sufficiently large, then we can double the size of training set at each iteration and use BFGS method to solve ERM subproblems, while we ensure the solution from previous round stays within the superlinear convergence neighborhood of the next problem. As a result, each subproblem can be solved with at most three BFGS updates with step size $1$. In comparison with Ada Newton, AdaQN has three major advantages. (I) Ada Newton requires a backtracking technique to determine the growth factor, while for AdaQN we can double the size of training set if the size of initial set is sufficiently large. (II) Ada Newton implementation requires computing the Hessian inverse at each iteration, while AdaQN only requires computing a single matrix inversion. (III) Ada Newton requires $2N$ Hessian computations overall, while AdaQN only needs $m_0$ Hessian evaluations at the initialization step.

In this paper, we showed that using quasi-Newton methods we only require constant number of iterations that is independent of the condition number $\kappa$ and the dimension $d$ to solve each subproblem to its statistical accuracy. More specifically, as the superlinear convergence rate of $(1/\sqrt{t})^t$ for BFGS is independent of the problem parameters, each subproblem can be solved with at most three iterations. On the other hand, any other second-order method that enjoys a local convergence rate (either linear or superlinear) that is independent of the problem parameters could be capable of solving each subproblem to its statistical accuracy with a finite constant number of iterations that is independent of the condition number $\kappa$ and dimension $d$. Hence, studying the application of such second-order or quasi-Newton methods for the considered adaptive sample size scheme is an interesting direction of research that requires further investigation.

One limitation of our results is that AdaQN provably outperforms other benchmarks only in the regime that $N \gg \max\left\{d, \kappa^2 s \log d\right\}$. This is due to the lower bound on the size of initial training set $m_0$. Notice that in many large-scale learning problems where $N$ is extremely large, this condition usually holds. We believe this limitation of our analysis could be resolved by tighter non-asymptotic analysis of BFGS method. Moreover, our guarantees only hold for convex settings and extension of our results to nonconvex settings for achieving first-order or second-order stationary points is a natural future direction.

## Acknowledgement

This research is supported in part by NSF Grant 2007668, ARO Grant W911NF2110226, the Machine Learning Laboratory at UT Austin, and the NSF AI Institute for Foundations of Machine Learning.

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
