# Appendix

## A    Further details of the proposed AdaQN method

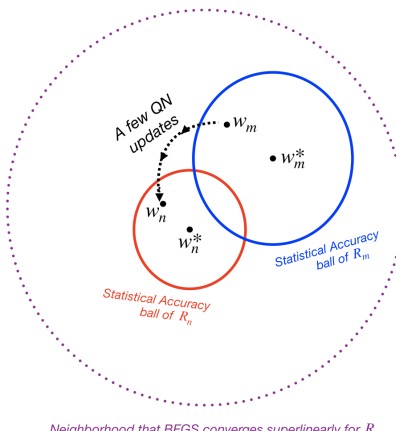

Figure 5: The interplay between superlinear convergence of BFGS and statistical accuracy in AdaQN.

In this section, we provide further intuition about the proposed AdaQN method. As shown in Figure 5, in AdaQN we need to ensure that the approximate solution of the ERM problem with $m$ samples denoted by $\mathbf{w}_m$ is within the superlinear convergence neighborhood of BFGS for the ERM problem with $n = 2m$ samples. Here, $\mathbf{w}_m^*$ and $\mathbf{w}_n^*$ are the optimal solutions of the risks $R_m$ and $R_n$ corresponding to the sets $\mathcal{S}_m$ and $\mathcal{S}_n$ with $m$ and $n$ samples, respectively, where $\mathcal{S}_m \subset \mathcal{S}_n$. The statistical accuracy region of $R_m$ is denoted by a blue circle, the statistical accuracy region of $R_n$ is denoted by a red circle, and the superlinear convergence neighborhood of BFGS for $R_n$ is denoted by a dotted purple circle. As we observe, any point within the statistical accuracy of $\mathbf{w}_m^*$ is within the superlinear convergence neighborhood of BFGS for $R_n$. Therefore, after a few steps (at most three steps) of BFGS, we find a new solution $\mathbf{w}_n$ that is within the statistical accuracy of $R_n$.

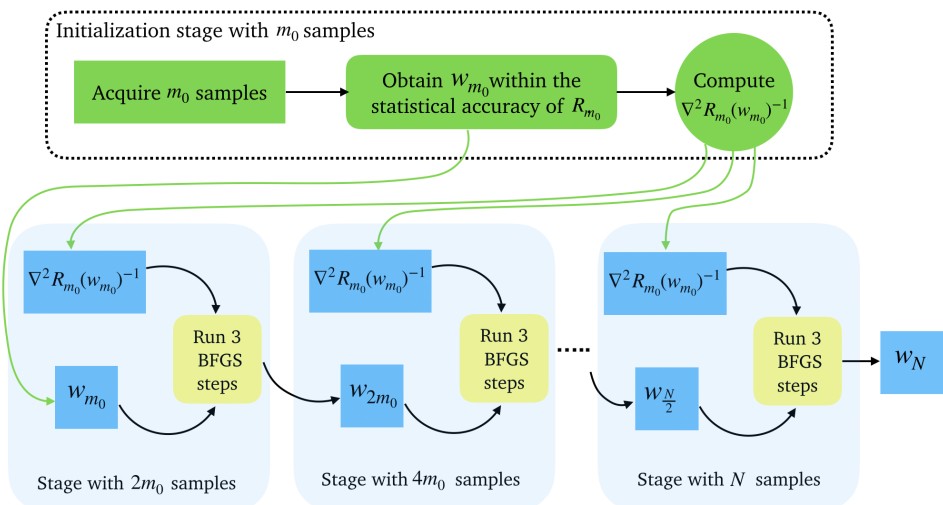

Figure 6: The phases of AdaQN.

In Figure 6, we illustrate the sequential steps of AdaQN moving from one stage to another stage. Note that in the initialization step we solve the first ERM problem with $m_0$ up to its statistical accuracy

to find an approximate solution $\mathbf{w}_{m_0}$. Then, we compute the Hessian $\nabla^2 R_{m_0}(\mathbf{w}_{m_0})$ and its inverse $\nabla^2 R_{m_0}(\mathbf{w}_{m_0})^{-1}$. Once the initialization step is done, we perform BFGS updates on the loss of ERM problem with $2m_0$ samples starting from the point $\mathbf{w}_{m_0}$ and using the initial Hessian inverse approximation $\nabla^2 R_{m_0}(\mathbf{w}_{m_0})^{-1}$. Then, after three BFGS updates we find $\mathbf{w}_{2m_0}$ that is within the statistical accuracy of $R_{2m_0}$. In the next stage, with $4m_0$ samples, we use the original Hessian inverse approximation $\nabla^2 R_{m_0}(\mathbf{w}_{m_0})^{-1}$ and the new variable $\mathbf{w}_{2m_0}$ for the BFGS updates. We keep doing this procedure until the size of the training set becomes $N$. As we observe, we only perform $m_0$ Hessian computations to find $\nabla^2 R_{m_0}(\mathbf{w}_{m_0})$ and one matrix inversion to find its inverse $\nabla^2 R_{m_0}(\mathbf{w}_{m_0})^{-1}$.

## B Proof of propositions and the main theorem

We start the proof by providing the following lemmas. These lemmas play fundamental roles in the proof of all our propositions and Theorem 1.

**Lemma 1.** *Matrix Bernstein Inequality. Consider a finite sequence $\{S_k\}_{k=1}^n$ of independent random symmetric matrix of dimension $d$. Suppose that $\mathbb{E}[S_k] = 0$ and $\|S_k\| \leq B$ for all $k$. Define that $Z = \sum_{k=1}^n S_k$ and we can bound the expectation,*

$$\mathbb{E}[\|Z\|] \leq \sqrt{2\mathbb{V}[Z]\log d} + \frac{B}{3}\log d, \tag{14}$$

*where $\mathbb{V}[Z] = \|\mathbb{E}[Z^2]\|$ is the matrix variance statistic of $Z$.*

*Proof.* Check Theorem 6.6.1 of [36]. $\qquad\square$

**Lemma 2.** *Suppose Assumptions 1-2 hold and the sample size $n = \Omega(\log d)$. Then the expectation of the difference between Hessian of the empirical risk $R_n$ and expected risk $R$ is bounded above by $\mathcal{O}\left(L\sqrt{\frac{s\log d}{n}}\right)$ in terms of the Frobenius norm for any $\mathbf{w}$, i.e,*

$$\mathbb{E}\left[\|\nabla^2 R_n(\mathbf{w}) - \nabla^2 R(\mathbf{w})\|_F\right] = \mathcal{O}\left(L\sqrt{\frac{s\log d}{n}}\right), \tag{15}$$

*where $s = \sup_{\mathbf{w},n}\left(\frac{\mathbb{E}[\|\nabla^2 R_n(\mathbf{w}) - \nabla^2 R(\mathbf{w})\|_F]}{\mathbb{E}[\|\nabla^2 R_n(\mathbf{w}) - \nabla^2 R(\mathbf{w})\|]}\right)^2$.*

*Proof.* We use Lemma 1 and define that $S_k = \frac{1}{n}[\nabla^2 f(\mathbf{w}; \mathbf{z}_k) - \nabla^2 R(\mathbf{w})]$ and thus $Z = \sum_{k=1}^n S_k = \nabla^2 R_n(\mathbf{w}) - \nabla^2 R(\mathbf{w})$. Notice that $\mathbb{E}[S_k] = 0$ and $\|S_k\| \leq \frac{2L}{n}$ for all $k$ so we know that $B = \frac{2L}{n}$. Since each sample is drawn independently we get

$$\mathbb{V}[Z] = \|\sum_{k=1}^n \mathbb{E}[S_k^2]\| \leq \sum_{k=1}^n \|\mathbb{E}[S_k^2]\| \leq \sum_{k=1}^n \mathbb{E}[\|S_k\|^2] \leq \sum_{k=1}^n \frac{4L^2}{n^2} = \frac{4L^2}{n}.$$

By the Matrix Bernstein Inequality of (14) we know,

$$\mathbb{E}\left[\|\nabla^2 R_n(\mathbf{w}) - \nabla^2 R(\mathbf{w})\|\right] \leq \sqrt{\frac{8L^2\log d}{n}} + \frac{2L}{3n}\log d = (2\sqrt{2}L + \frac{2L}{3}\sqrt{\frac{\log d}{n}})\sqrt{\frac{\log d}{n}}.$$

Notice that the sample size $n = \Omega(\log d)$, thus we have that

$$2\sqrt{2}L + \frac{2L}{3}\sqrt{\frac{\log d}{n}} = \mathcal{O}(L),$$

$$\mathbb{E}\left[\|\nabla^2 R_n(\mathbf{w}) - \nabla^2 R(\mathbf{w})\|\right] = \mathcal{O}\left(L\sqrt{\frac{\log d}{n}}\right).$$

By the definition of $s = \sup_{\mathbf{w},n}\left(\frac{\mathbb{E}[\|\nabla^2 R_n(\mathbf{w}) - \nabla^2 R(\mathbf{w})\|_F]}{\mathbb{E}[\|\nabla^2 R_n(\mathbf{w}) - \nabla^2 R(\mathbf{w})\|]}\right)^2$ we know

$$\mathbb{E}\left[\|\nabla^2 R_n(\mathbf{w}) - \nabla^2 R(\mathbf{w})\|_F\right] \leq \sqrt{s}\mathbb{E}\left[\|\nabla^2 R_n(\mathbf{w}) - \nabla^2 R(\mathbf{w})\|\right],$$

thus we have

$$\mathbb{E}\left[\|\nabla^2 R_n(\mathbf{w}) - \nabla^2 R(\mathbf{w})\|_F\right] = \mathcal{O}\left(L\sqrt{\frac{s\log d}{n}}\right).$$

This is correct for all $\mathbf{w}$ so we prove the conclusion (15). □

**Lemma 3.** *Suppose $S_m$ and $S_n$ are two datasets and $S_m \subset S_n \subset \mathcal{T}$. Assume that there are $m$ samples in $S_m$ and $n$ samples in $S_n$ and $n \geq m = \Omega(\log d)$. Consider the corresponding empirical risk loss function $R_m$ and $R_n$ defined on $S_m$ and $S_n$, respectively. Then for any $\mathbf{w}$ the expectation of the difference between their Hessians is bounded above by*

$$\mathbb{E}\left[\|\nabla^2 R_n(\mathbf{w}) - \nabla^2 R_m(\mathbf{w})\|_F\right] = \mathcal{O}\left(L\sqrt{\frac{s\log d}{m}}\right), \tag{16}$$

*where $s = \sup_{\mathbf{w},n}\left(\frac{\mathbb{E}\left[\|\nabla^2 R_n(\mathbf{w}) - \nabla^2 R(\mathbf{w})\|_F\right]}{\mathbb{E}\left[\|\nabla^2 R_n(\mathbf{w}) - \nabla^2 R(\mathbf{w})\|\right]}\right)^2.$*

*Proof.* Using triangle inequality we have that

$$\|\nabla^2 R_n(\mathbf{w}) - \nabla^2 R_m(\mathbf{w})\|_F \leq \|\nabla^2 R_n(\mathbf{w}) - \nabla^2 R(\mathbf{w})\|_F + \|\nabla^2 R_m(\mathbf{w}) - \nabla^2 R(\mathbf{w})\|_F.$$

By (15) of Lemma 2 and $n \geq m = \Omega(\log d)$, we have

$$\mathbb{E}\left[\|\nabla^2 R_m(\mathbf{w}) - \nabla^2 R(\mathbf{w})\|_F\right] = \mathcal{O}\left(L\sqrt{\frac{s\log d}{m}}\right).$$

$$\mathbb{E}\left[\|\nabla^2 R_n(\mathbf{w}) - \nabla^2 R(\mathbf{w})\|_F\right] = \mathcal{O}\left(L\sqrt{\frac{s\log d}{n}}\right) = \mathcal{O}\left(L\sqrt{\frac{s\log d}{m}}\right).$$

Leveraging the above three inequalities we prove the conclusion (16). □

**Lemma 4.** *Assume that $f(x)$ is a strictly convex and self-concordant function. Suppose that $x, y \in dom(f)$, then we have*

$$f(y) \geq f(x) + \nabla f(x)^\top(y - x) + w(\|\nabla^2 f(x)^{\frac{1}{2}}(y - x)\|), \tag{17}$$

*where $w(t) := t - \ln(1 + t)$ and for $t \geq 0$ we have*

$$w(t) \geq \frac{t^2}{2(1 + t)}. \tag{18}$$

*Moreover, if $x, y \in dom(f)$ satisfy that $\|\nabla^2 f(x)^{\frac{1}{2}}(y - x)\| \leq \frac{1}{2}$, then we have:*

$$\frac{1}{1 + 6\|\nabla^2 f(x)^{\frac{1}{2}}(y - x)\|}\nabla^2 f(x) \preceq \nabla^2 f(y) \preceq (1 + 6\|\nabla^2 f(x)^{\frac{1}{2}}(y - x)\|)\nabla^2 f(x). \tag{19}$$

*Proof.* Check Theorem 4.1.7 and Lemma 5.1.5 of [37] for (17) and (18). If $x, y \in dom(f)$ satisfy that $\|\nabla^2 f(x)^{\frac{1}{2}}(y - x)\| \leq \frac{1}{2} < 1$, by Theorem 4.1.6 of [37] we obtain that

$$(1 - \|\nabla^2 f(x)^{\frac{1}{2}}(y - x)\|)^2 \nabla^2 f(x) \preceq \nabla^2 f(y) \preceq \frac{1}{(1 - \|\nabla^2 f(x)^{\frac{1}{2}}(y - x)\|)^2}\nabla^2 f(x). \tag{20}$$

Notice that for $a \in [0, \frac{1}{2}]$ we have

$$\frac{1}{(1 - a)^2} = 1 + \frac{2 - a}{(1 - a)^2}a \leq 1 + 6a. \tag{21}$$

Combining (20) and (21) we can obtain that (19) holds. □

**Lemma 5.** *Suppose $t > 0$ satisfies the following condition*

$$t \geq C_1 + \ln \frac{1}{C_2}, \tag{22}$$

*where $C_1$ and $C_2$ are two positive constants. Then we can obtain the following inequality*

$$\left( \frac{C_1}{t} \right)^t \leq C_2. \tag{23}$$

*Proof.* Denote $s = \frac{t}{C_1}$ and based on (23) we have the following inequality

$$\left( \frac{1}{s} \right)^{C_1 s} \leq C_2.$$

This is equivalent to

$$s^{C_1 s} \geq \frac{1}{C_2}.$$

Take the nature logarithm on both sides we get

$$C_1 s \ln s \geq \ln \frac{1}{C_2}. \tag{24}$$

This shows that condition (24) is equivalent to condition (23). Notice that for any $x > 0$ we have the following inequality

$$\ln x \geq 1 - \frac{1}{x}.$$

So if $s$ satisfies

$$C_1 s (1 - \frac{1}{s}) \geq \ln \frac{1}{C_2}, \tag{25}$$

we can derive that

$$C_1 s \ln s \geq C_1 s (1 - \frac{1}{s}) \geq \ln \frac{1}{C_2}.$$

And since $t = C_1 s$ the condition (25) is equivalent to

$$C_1 s - C_1 \geq \ln \frac{1}{C_2},$$

$$t \geq C_1 + \ln \frac{1}{C_2}.$$

This is just the condition (22). So if $t$ satisfies the condition (22) which is equivalent to condition (25), condition (24) will be satisfied which is equivalent to condition (23). □

Now we present the proof of Propositions 1-4.

### B.1   Proof of Proposition 1

Note that the difference $R_n(\mathbf{w}_m) - R_n(\mathbf{w}_n^*)$ can be written as

$$R_n(\mathbf{w}_m) - R_n(\mathbf{w}_n^*) = R_n(\mathbf{w}_m) - R_m(\mathbf{w}_m) + R_m(\mathbf{w}_m) - R_m(\mathbf{w}_m^*) \\ + R_m(\mathbf{w}_m^*) - R(\mathbf{w}^*) + R(\mathbf{w}^*) - R_n(\mathbf{w}_n^*). \tag{26}$$

Notice that all samples are drawn independently in our algorithm and all the expectations are with respect to the corresponding training set. Therefore, for the training sets $S_m, S_n$ with $m < n$, where $S_m \subset S_n$, and their corresponding objective functions are $R_m(\mathbf{w})$ and $R_n(\mathbf{w})$, we have

$$\mathbb{E}_{S_n}[R_n(\mathbf{w})] = R(\mathbf{w}),$$

$$\mathbb{E}_{S_n}[R_m(\mathbf{w})] = \mathbb{E}_{S_{n-m}}[\mathbb{E}_{S_m}[R_m(\mathbf{w})]] = \mathbb{E}_{S_m}[R_m(\mathbf{w})] = R(\mathbf{w}),$$

where $R(\mathbf{w})$ is the expected loss function. Computing the expectation of both sides in (26) and dismissing the notations of the corresponding training set implies that

$$\mathbb{E}[R_n(\mathbf{w}_m) - R_n(\mathbf{w}_n^*)] = \mathbb{E}[R_n(\mathbf{w}_m) - R_m(\mathbf{w}_m)] + \mathbb{E}[R_m(\mathbf{w}_m) - R_m(\mathbf{w}_m^*)] \\ + \mathbb{E}[R_m(\mathbf{w}_m^*) - R(\mathbf{w}^*)] + \mathbb{E}[R(\mathbf{w}^*) - R_n(\mathbf{w}_n^*)]. \tag{27}$$

Now note that
$$\mathbb{E}\left[R_n(\mathbf{w}_m) - R_m(\mathbf{w}_m)\right] = R(\mathbf{w}_m) - R(\mathbf{w}_m) = 0.$$

Moreover, since $\mathbf{w}_m$ solves the ERM problem $R_m$ within its statistical accuracy, we have
$$\mathbb{E}\left[R_m(\mathbf{w}_m) - R_m(\mathbf{w}_m^*)\right] \leq V_m.$$

Using (3) and the fact that $n \geq m$ we can show that
$$\mathbb{E}\left[R_m(\mathbf{w}_m^*) - R(\mathbf{w}^*)\right] \leq \mathbb{E}\left[|R_m(\mathbf{w}_m^*) - R(\mathbf{w}^*)|\right] \leq V_m,$$

$$\mathbb{E}\left[R(\mathbf{w}^*) - R_n(\mathbf{w}_n^*)\right] \leq \mathbb{E}\left[|R_n(\mathbf{w}_n^*) - R(\mathbf{w}^*)|\right] \leq V_n \leq V_m.$$

Leveraging all the above inequalities and replacing the terms in (27) by their upper bounds implies
$$\mathbb{E}[R_n(\mathbf{w}_m) - R_n(\mathbf{w}_n^*)] \leq 3V_m$$

and the claim in (7) of Proposition 1 follows.

## B.2 Proof of Proposition 2

Check Corollary 5.5 of [24].

## B.3 Proof of Proposition 3

Recall that $R_n$ is strongly convex with $\mu$ and its gradient is smooth with $L$, Hence, we have

$$\|\nabla^2 R_n(\mathbf{w}_n^*)^{\frac{1}{2}}(\mathbf{w}_m - \mathbf{w}_n^*)\| \leq \|\nabla^2 R_n(\mathbf{w}_n^*)\|^{\frac{1}{2}}\|\mathbf{w}_m - \mathbf{w}_n^*\| \leq \sqrt{\frac{2L}{\mu}\left[R_n(\mathbf{w}_m) - R_n(\mathbf{w}_n^*)\right]}.$$

By Proposition 1 we know that
$$\mathbb{E}\left[R_n(\mathbf{w}_m) - R_n(\mathbf{w}_n^*)\right] \leq 3V_m.$$

Using Jensen's inequality (notice that function $f(x) = \sqrt{x}$ is concave) we obtain

$$\mathbb{E}\left[\|\nabla^2 R_n(\mathbf{w}_n^*)^{\frac{1}{2}}(\mathbf{w}_m - \mathbf{w}_n^*)\|\right] \leq \sqrt{\frac{2L}{\mu}\mathbb{E}\left[R_n(\mathbf{w}_m) - R_n(\mathbf{w}_n^*)\right]} \leq \sqrt{6\kappa V_m}.$$

Moreover, we assume that $V_m = \mathcal{O}(\frac{1}{m})$ under the condition that $m = \Omega(\kappa^2 \log d)$. So when $m$ is lower bounded by

$$m = \Omega\left(\max\{\frac{6\kappa}{(\frac{1}{300})^2}, \kappa^2 \log d\}\right) = \Omega\left(\max\{\kappa, \kappa^2 \log d\}\right) = \Omega(\kappa^2 \log d),$$

we can ensure that

$$\mathbb{E}\left[\|\nabla^2 R_n(\mathbf{w}_n^*)^{\frac{1}{2}}(\mathbf{w}_m - \mathbf{w}_n^*)\|\right] \leq \sqrt{6\kappa V_m} \leq \frac{1}{300}.$$

## B.4 Proof of Proposition 4

Notice that by triangle inequality we have

$$
\begin{aligned}
&\|\nabla^2 R_n(\mathbf{w}_n^*)^{-\frac{1}{2}}\left[\nabla^2 R_{m_0}(\mathbf{w}_{m_0}) - \nabla^2 R_n(\mathbf{w}_n^*)\right]\nabla^2 R_n(\mathbf{w}_n^*)^{-\frac{1}{2}}\|_F \\
\leq &\|\nabla^2 R_n(\mathbf{w}_n^*)^{-\frac{1}{2}}\left[\nabla^2 R_{m_0}(\mathbf{w}_{m_0}) - \nabla^2 R_n(\mathbf{w}_{m_0})\right]\nabla^2 R_n(\mathbf{w}_n^*)^{-\frac{1}{2}}\|_F \quad + \\
&\|\nabla^2 R_n(\mathbf{w}_n^*)^{-\frac{1}{2}}\left[\nabla^2 R_n(\mathbf{w}_{m_0}) - \nabla^2 R_n(\mathbf{w}_n^*)\right]\nabla^2 R_n(\mathbf{w}_n^*)^{-\frac{1}{2}}\|_F.
\end{aligned}
\tag{28}
$$

Recall that $R_n$ is strongly convex with $\mu$ and thus $\|\nabla^2 R_n(\mathbf{w}_n^*)^{-\frac{1}{2}}\| \leq \sqrt{\frac{1}{\mu}}$. By (16) of Lemma 3 we derive that when $m_0 = \Omega(\log d)$,

$$\mathbb{E}\left[\|\nabla^2 R_n(\mathbf{w}_n^*)^{-\frac{1}{2}}\left[\nabla^2 R_{m_0}(\mathbf{w}_{m_0}) - \nabla^2 R_n(\mathbf{w}_{m_0})\right]\nabla^2 R_n(\mathbf{w}_n^*)^{-\frac{1}{2}}\|_F\right]$$

$$\leq \mathbb{E}\left[\|\nabla^2 R_n(\mathbf{w}_n^*)^{-\frac{1}{2}}\|^2 \|\nabla^2 R_{m_0}(\mathbf{w}_{m_0}) - \nabla^2 R_n(\mathbf{w}_{m_0})\|_F\right]$$

$$\leq \frac{1}{\mu}\mathbb{E}\left[\|\nabla^2 R_n(\mathbf{w}_{m_0}) - \nabla^2 R_{m_0}(\mathbf{w}_{m_0})\|_F\right]$$

$$= \mathcal{O}\left(\frac{L}{\mu}\sqrt{\frac{s\log d}{m_0}}\right)$$

$$= \mathcal{O}\left(\kappa\sqrt{\frac{s\log d}{m_0}}\right).$$

So when $m_0$ is lower bounded by

$$m_0 = \Omega\left(\max\{\frac{\kappa^2 s\log d}{(\frac{1}{14})^2}, \log d\}\right) = \Omega\left(\max\{\kappa^2 s\log d, \log d\}\right) = \Omega(\kappa^2 s\log d), \quad (29)$$

we can ensure that,

$$\mathbb{E}\left[\|\nabla^2 R_n(\mathbf{w}_n^*)^{-\frac{1}{2}}\left[\nabla^2 R_{m_0}(\mathbf{w}_{m_0}) - \nabla^2 R_n(\mathbf{w}_{m_0})\right]\nabla^2 R_n(\mathbf{w}_n^*)^{-\frac{1}{2}}\|_F\right] = \mathcal{O}\left(\kappa\sqrt{\frac{s\log d}{m_0}}\right) \leq \frac{1}{14}.$$

$$(30)$$

Using the same techniques in the proof of Proposition 3 we know that when $m_0$ is lower bounded by

$$m_0 = \Omega(\kappa^2 \log d), \quad (31)$$

we can ensure that

$$\mathbb{E}\left[\|\nabla^2 R_n(\mathbf{w}_n^*)^{\frac{1}{2}}(\mathbf{w}_{m_0} - \mathbf{w}_n^*)\|\right] \leq \frac{1}{300}.$$

By Markov's inequality we know that

$$\mathbb{P}\left(\|\nabla^2 R_n(\mathbf{w}_n^*)^{\frac{1}{2}}(\mathbf{w}_{m_0} - \mathbf{w}_n^*)\| \leq \frac{1}{2}\right)$$

$$= 1 - \mathbb{P}\left(\|\nabla^2 R_n(\mathbf{w}_n^*)^{\frac{1}{2}}(\mathbf{w}_{m_0} - \mathbf{w}_n^*)\| \geq \frac{1}{2}\right)$$

$$\geq 1 - \frac{\mathbb{E}\left[\|\nabla^2 R_n(\mathbf{w}_n^*)^{\frac{1}{2}}(\mathbf{w}_{m_0} - \mathbf{w}_n^*)\|\right]}{1/2}$$

$$\geq 1 - \frac{1/300}{1/2} = \frac{149}{150}.$$

This indicates that with high probability(w.h.p) of at least $149/150$ we get that

$$\|\nabla^2 R_n(\mathbf{w}_n^*)^{\frac{1}{2}}(\mathbf{w}_{m_0} - \mathbf{w}_n^*)\| \leq \frac{1}{2}.$$

Using (19) of Lemma 4 we have that with high probability

$$\frac{\nabla^2 R_n(\mathbf{w}_n^*)}{1 + 6\|\nabla^2 R_n(\mathbf{w}_n^*)^{\frac{1}{2}}(\mathbf{w}_{m_0} - \mathbf{w}_n^*)\|} \preceq \nabla^2 R_n(\mathbf{w}_{m_0}) \preceq (1 + 6\|\nabla^2 R_n(\mathbf{w}_n^*)^{\frac{1}{2}}(\mathbf{w}_{m_0} - \mathbf{w}_n^*)\|)\nabla^2 R_n(\mathbf{w}_n^*).$$

Times the matrix $\nabla^2 R_n(\mathbf{w}_n^*)^{-\frac{1}{2}}$ on both sides we get that

$$\frac{1}{1 + 6\|\nabla^2 R_n(\mathbf{w}_n^*)^{\frac{1}{2}}(\mathbf{w}_{m_0} - \mathbf{w}_n^*)\|}I \preceq \nabla^2 R_n(\mathbf{w}_n^*)^{-\frac{1}{2}}\nabla^2 R_n(\mathbf{w}_{m_0})\nabla^2 R_n(\mathbf{w}_n^*)^{-\frac{1}{2}}, \quad w.h.p,$$

$$\nabla^2 R_n(\mathbf{w}_n^*)^{-\frac{1}{2}}\nabla^2 R_n(\mathbf{w}_{m_0})\nabla^2 R_n(\mathbf{w}_n^*)^{-\frac{1}{2}} \preceq (1 + 6\|\nabla^2 R_n(\mathbf{w}_n^*)^{\frac{1}{2}}(\mathbf{w}_{m_0} - \mathbf{w}_n^*)\|)I, \quad w.h.p.$$

So we achieve that

$$\|\nabla^2 R_n(\mathbf{w}_n^*)^{-\frac{1}{2}}[\nabla^2 R_n(\mathbf{w}_{m_0}) - \nabla^2 R_n(\mathbf{w}_n^*)]\nabla^2 R_n(\mathbf{w}_n^*)^{-\frac{1}{2}}\|$$

$$=\|\nabla^2 R_n(\mathbf{w}_n^*)^{-\frac{1}{2}}\nabla^2 R_n(\mathbf{w}_{m_0})\nabla^2 R_n(\mathbf{w}_n^*)^{-\frac{1}{2}} - I\|$$

$$\leq \max\{6\|\nabla^2 R_n(\mathbf{w}_n^*)^{\frac{1}{2}}(\mathbf{w}_{m_0} - \mathbf{w}_n^*)\|, 1 - \frac{1}{1 + 6\|\nabla^2 R_n(\mathbf{w}_n^*)^{\frac{1}{2}}(\mathbf{w}_{m_0} - \mathbf{w}_n^*)\|}\}$$

$$=6\|\nabla^2 R_n(\mathbf{w}_n^*)^{\frac{1}{2}}(\mathbf{w}_{m_0} - \mathbf{w}_n^*)\|, \quad w.h.p.$$

Hence we obtain that

$$\|\nabla^2 R_n(\mathbf{w}_n^*)^{-\frac{1}{2}}[\nabla^2 R_n(\mathbf{w}_{m_0}) - \nabla^2 R_n(\mathbf{w}_n^*)]\nabla^2 R_n(\mathbf{w}_n^*)^{-\frac{1}{2}}\|_F$$

$$\leq \sqrt{d}\|\nabla^2 R_n(\mathbf{w}_n^*)^{-\frac{1}{2}}[\nabla^2 R_n(\mathbf{w}_{m_0}) - \nabla^2 R_n(\mathbf{w}_n^*)]\nabla^2 R_n(\mathbf{w}_n^*)^{-\frac{1}{2}}\| \tag{32}$$

$$\leq 6\sqrt{d}\|\nabla^2 R_n(\mathbf{w}_n^*)^{\frac{1}{2}}(\mathbf{w}_{m_0} - \mathbf{w}_n^*)\|, \quad w.h.p,$$

where we use the fact that $\|A\|_F \leq \sqrt{d}\|A\|$ for any matrix $A \in \mathbb{R}^{d \times d}$. Because $\mathbf{w}_n^*$ is the optimal solution of the function $R_n(\mathbf{w})$ we know that $\nabla R_n(\mathbf{w}_n^*) = 0$. Recall that $w(t) = t - \ln(1 + t)$ and by (17) of Lemma 4 we get that

$$w(\|\nabla^2 R_n(\mathbf{w}_n^*)^{\frac{1}{2}}(\mathbf{w}_{m_0} - \mathbf{w}_n^*)\|) \leq R_n(\mathbf{w}_{m_0}) - R_n(\mathbf{w}_n^*). \tag{33}$$

By (18) of Lemma 4 and $\|\nabla^2 R_n(\mathbf{w}_n^*)^{\frac{1}{2}}(\mathbf{w}_{m_0} - \mathbf{w}_n^*)\| \leq \frac{1}{2}$ w.h.p, we obtain that with high probability

$$w(\|\nabla^2 R_n(\mathbf{w}_n^*)^{\frac{1}{2}}(\mathbf{w}_{m_0} - \mathbf{w}_n^*)\|) \geq \frac{\|\nabla^2 R_n(\mathbf{w}_n^*)^{\frac{1}{2}}(\mathbf{w}_{m_0} - \mathbf{w}_n^*)\|^2}{2(1 + \|\nabla^2 R_n(\mathbf{w}_n^*)^{\frac{1}{2}}(\mathbf{w}_{m_0} - \mathbf{w}_n^*)\|)} \geq \frac{\|\nabla^2 R_n(\mathbf{w}_n^*)^{\frac{1}{2}}(\mathbf{w}_{m_0} - \mathbf{w}_n^*)\|^2}{3}. \tag{34}$$

Combining (32), (33) and (34) we have that

$$\|\nabla^2 R_n(\mathbf{w}_n^*)^{-\frac{1}{2}}[\nabla^2 R_n(\mathbf{w}_{m_0}) - \nabla^2 R_n(\mathbf{w}_n^*)]\nabla^2 R_n(\mathbf{w}_n^*)^{-\frac{1}{2}}\|_F$$

$$\leq 6\sqrt{d}\|\nabla^2 R_n(\mathbf{w}_n^*)^{\frac{1}{2}}(\mathbf{w}_{m_0} - \mathbf{w}_n^*)\|$$

$$\leq 6\sqrt{3}\sqrt{d}\sqrt{w(\|\nabla^2 R_n(\mathbf{w}_n^*)^{\frac{1}{2}}(\mathbf{w}_{m_0} - \mathbf{w}_n^*)\|)}$$

$$\leq 6\sqrt{3}\sqrt{d}\sqrt{R_n(\mathbf{w}_{m_0}) - R_n(\mathbf{w}_n^*)}, \quad w.h.p.$$

By proposition 1 we know that

$$\mathbb{E}[R_n(\mathbf{w}_{m_0}) - R_n(\mathbf{w}_n^*)] \leq 3V_{m_0}.$$

Using Jensen's inequality (notice that function $f(x) = \sqrt{x}$ is concave) we get that

$$\mathbb{E}\left[\|\nabla^2 R_n(\mathbf{w}_n^*)^{-\frac{1}{2}}[\nabla^2 R_n(\mathbf{w}_{m_0}) - \nabla^2 R_n(\mathbf{w}_n^*)]\nabla^2 R_n(\mathbf{w}_n^*)^{-\frac{1}{2}}\|_F\right]$$

$$\leq 6\sqrt{3}\sqrt{d}\sqrt{\mathbb{E}[R_n(\mathbf{w}_{m_0}) - R_n(\mathbf{w}_n^*)]}$$

$$\leq 18\sqrt{dV_{m_0}}.$$

We assume that $V_{m_0} = \mathcal{O}(\frac{1}{m_0})$ under the condition that $m_0 = \Omega(\kappa^2 \log d)$. So when $m_0$ is lower bounded by

$$m_0 = \Omega\left(\max\{\frac{18^2 d}{(\frac{1}{14})^2}, \kappa^2 \log d\}\right) = \Omega\left(\max\{d, \kappa^2 \log d\}\right), \tag{35}$$

we can ensure that

$$\mathbb{E}\left[\|\nabla^2 R_n(\mathbf{w}_n^*)^{-\frac{1}{2}}[\nabla^2 R_n(\mathbf{w}_{m_0}) - \nabla^2 R_n(\mathbf{w}_n^*)]\nabla^2 R_n(\mathbf{w}_n^*)^{-\frac{1}{2}}\|_F\right] \leq 18\sqrt{dV_{m_0}} \leq \frac{1}{14}. \tag{36}$$

Leveraging (28), (29), (30), (31), (35) and (36) and using the fact that $s \geq 1$ we know that when the initial sample size $m_0$ is lower bounded by

$$m_0 = \Omega\left(\max\{d, \kappa^2 \log d, \kappa^2 s \log d\}\right) = \Omega\left(\max\{d, \kappa^2 s \log d\}\right),$$

we have that

$$\mathbb{E}\left[\|\nabla^2 R_n(\mathbf{w}_n^*)^{-\frac{1}{2}} \left[\nabla^2 R_{m_0}(\mathbf{w}_{m_0}) - \nabla^2 R_n(\mathbf{w}_n^*)\right] \nabla^2 R_n(\mathbf{w}_n^*)^{-\frac{1}{2}}\|_F\right]$$

$$\leq \mathbb{E}\left[\|\nabla^2 R_n(\mathbf{w}_n^*)^{-\frac{1}{2}} \left[\nabla^2 R_{m_0}(\mathbf{w}_{m_0}) - \nabla^2 R_n(\mathbf{w}_{m_0})\right] \nabla^2 R_n(\mathbf{w}_n^*)^{-\frac{1}{2}}\|_F\right] \quad +$$

$$\mathbb{E}\left[\|\nabla^2 R_n(\mathbf{w}_n^*)^{-\frac{1}{2}} \left[\nabla^2 R_n(\mathbf{w}_{m_0}) - \nabla^2 R_n(\mathbf{w}_n^*)\right] \nabla^2 R_n(\mathbf{w}_n^*)^{-\frac{1}{2}}\|_F\right]$$

$$\leq \frac{1}{14} + \frac{1}{14}$$

$$= \frac{1}{7}.$$

So the proof of Proposition 4 is complete.

Finally we present the proof of the main theorem of the paper.

## B.5  Proof of Theorem 1

Suppose we are at the stage with $n = 2m$ samples and the initial iterate $\mathbf{w}_m$ is within the statistical accuracy of $R_m$. Further, suppose the size of initial set $m_0$ satisfies (12) and by $s \geq 1$ we have

$$m_0 = \Omega(\kappa^2 \log d),$$

hence by Proposition 3 and 4 the conditions in (8) are satisfied in expectation. If we define $\mathbf{w}_n$ as the output of the process after running $t_n$ updates of BFGS on $R_n$ with step size 1, then Proposition 2 implies that

$$\mathbb{E}[R_n(\mathbf{w}_n) - R_n(\mathbf{w}_n^*)] \leq 1.1 \ (1/t_n)^{t_n} \ \mathbb{E}[R_n(\mathbf{w}_m) - R_n(\mathbf{w}_n^*)].$$

Moreover, if we set $n = 2m$ by Proposition 1 we obtain $\mathbb{E}[R_n(\mathbf{w}_m) - R_n(\mathbf{w}_n^*)] \leq 3V_m$. Combining these two inequalities implies that

$$\mathbb{E}[R_n(\mathbf{w}_n) - R_n(\mathbf{w}_n^*)] \leq 3.3 \ V_m \ (1/t_n)^{t_n} . \tag{37}$$

Our goal is to ensure that the output iterate $\mathbf{w}_n$ reaches the statistical accuracy of $R_n$ and satisfies $\mathbb{E}[R_n(\mathbf{w}_n) - R_n(\mathbf{w}_n^*)] \leq V_n$. This condition is indeed satisfied if the upper bound in (37) is smaller than $V_n$ and we have

$$3.3 \ V_m \ (1/t_n)^{t_n} \leq V_n. \tag{38}$$

As $V_n = \mathcal{O}(1/n)$ (since $n \geq m_0 = \Omega(\kappa^2 \log d)$) and $n = 2m$, condition (38) is equivalent to $(1/t_n)^{t_n} \leq (1/6.6)$. It can be easily verified that this condition holds if $t_n$ satisfies (see Lemma 5):

$$t_n \geq 1 + \ln (6.6). \tag{39}$$

The expression in (39) shows that, in the stage that the number of active samples is $n$, we need to run BFGS for at most $t_n = 3$ iterations (since $3 \geq 1 + \ln(6.6)$). Note that this threshold is independent of $n$ or any other parameters. Hence, the cost of running BFGS updates at the phase that we have $n$ samples in the training set is

$$n t_n \tau_{grad} + t_n \tau_{prod} = 3(n \tau_{grad} + \tau_{prod}),$$

where the first term corresponds to the cost of computing $n$ gradients for $t_n$ iterations and the second term corresponds to the cost of computing $t_n$ matrix-vector products for updating the Hessian inverse approximation. To compute the overall cost of AdaQN, we must sum up the cost of each phase from the initial training set with $m_0$ samples up to the last phase that we have $N$ samples in the active training set. For simplicity we assume that the total number of samples $N$ and the initial sample set $m_0$ satisfy the condition $N = 2^q m_0$ where $q \in \mathbb{Z}$. Hence, the computational cost of solving ERM problems from $m_0$ samples to $N$ samples is

$$\sum_{k=0}^{q} [3(2^k m_0 \tau_{grad} + \tau_{prod})]$$

$$\leq 3\left[\left(m_0 \tau_{grad} 2^{q+1}\right) + (q+1)\tau_{prod}\right] = 6N\tau_{grad} + 3\left(1 + \log\left(\frac{N}{m_0}\right)\right)\tau_{prod}$$

Further, we need to compute $m_0$ Hessians and one matrix inversion at the first stage, when we compute $\mathbf{H}_{m_0} = \nabla^2 R_{m_0}(\mathbf{w}_{m_0})^{-1}$. By combining these costs, the claim in Theorem 1 follows.

## C  Analysis of parameter s

As we discussed in the paper, the lower bound on the size of initial training set $m_0$, depends on the parameter $s$ which is formally defined as

$$s = \sup_{\mathbf{w},n} \left( \frac{\mathbb{E}\left[\|\nabla^2 R_n(\mathbf{w}) - \nabla^2 R(\mathbf{w})\|_F\right]}{\mathbb{E}\left[\|\nabla^2 R_n(\mathbf{w}) - \nabla^2 R(\mathbf{w})\|\right]} \right)^2.$$

This parameter could be as small as $1$ in the best case scenario, and as large as $d$ in the worst case scenario. This parameter depends heavily on the variation/variance of the input features for linear models. To better illustrate this point, we consider a simple case where the input vectors are $\mathbf{x} = [x_1, \ldots, x_d]$ and we use a linear regression model for our prediction function and a quadratic loss function.[2] In this case, the empirical risk $R_n(\mathbf{w})$ defined by the samples $\{\mathbf{x}^{(k)}, y^{(k)}\}_{k=1}^n$ and the expected risk are given by

$$R_n(\mathbf{w}) = \frac{1}{2n}\sum_{k=1}^n (y^{(k)} - \mathbf{w}^\top \mathbf{x}^{(k)})^2, \qquad R(\mathbf{w}) = \frac{1}{2}\mathbb{E}_{(\mathbf{x},y)}[(y - \mathbf{w}^\top \mathbf{x})^2].$$

And the Hessians $\nabla^2 R_n(\mathbf{w})$ and $\nabla^2 R(\mathbf{w})$ are given by

$$\nabla^2 R_n(\mathbf{w}) = \frac{1}{n}\sum_{k=1}^n \mathbf{x}^{(k)}\mathbf{x}^{(k)\top}, \qquad \nabla^2 R(\mathbf{w}) = \mathbb{E}_{(\mathbf{x},y)}[\mathbf{x}\mathbf{x}^\top].$$

To understand the behavior of the ratio $s$ we need to study the gap between these two matrices. Note that the $(i,j)$ element of these matrices are given by

$$\nabla^2 R_n(\mathbf{w})[i,j] = \frac{1}{n}\sum_{k=1}^n x_i^{(k)} x_j^{(k)}, \qquad \nabla^2 R(\mathbf{w})[i,j] = \mathbb{E}[x_i x_j],$$

where $x_i^{(k)}$ denotes the $i$-th element of the $k$-th sample point. Without loss of generality we assume that the correlation between two different features, i.e., $\mathbb{E}[x_i x_j]$ for $i \neq j$, is much smaller than the second moment of each feature $\mathbb{E}[x_i^2]$. Thus, we can focus on the diagonal components of these two matrices only. In that case, the $i$-th component on the diagonal of the difference matrix $\nabla^2 R_n(\mathbf{w}) - \nabla^2 R(\mathbf{w})$ is given by

$$e_i := \frac{1}{n}\sum_{k=1}^n (x_i^{(k)})^2 - \mathbb{E}[x_i^2],$$

which is equivalent to the concentration error of the random variable $x_i^2$. By the definition of Frobenius norm and $l_2$ norm we know that $s$ is approximately

$$s \approx \frac{\sum_{i=1}^d e_i^2}{\max_{i=1,\ldots,d} e_i^2}.$$

Now if different features of the input vector $\mathbf{x}$ have similar range and second moments, we would expect the parameters $e_i$ to be close to each other. In this case, the Frobenius norm of the difference matrix $\nabla^2 R_n(\mathbf{w}) - \nabla^2 R(\mathbf{w})$ is almost $\sqrt{d}$ of its operator norm and hence $s \approx d$. However, if the variance and range of different elements of the feature vector $\mathbf{x}$ are substantially different from each other, $s$ would be much smaller than $d$. For instance, when all the $e_i^2$ coordinate in the same range, except that $\max_i e_i^2$ is much larger than the rest then the Frobenius norm and operator norm of the difference matrix $\nabla^2 R_n(\mathbf{w}) - \nabla^2 R(\mathbf{w})$ are very close to each other and therefore $s = O(1)$. In this case, $s$ does not scale with $d$. For many datasets, the second scenario holds, since we often observe that the matrix of difference $\nabla^2 R_n(\mathbf{w}) - \nabla^2 R(\mathbf{w})$ has several eigenvalues with small absolute value and a few eigenvalues with large absolute value.

To better quantify the parameter $s$, we conduct the following numerical experiment. We generate the feature vector randomly $\mathbf{x} = [x_1, \ldots, x_d]$ where $x_i$ are independent random variables from normal distribution. We generate $n = 1000$ samples to compute the empirical risk Hessian $\nabla^2 R_n(\mathbf{w}) =$

---

[2]Case of the logistic regression is similar.

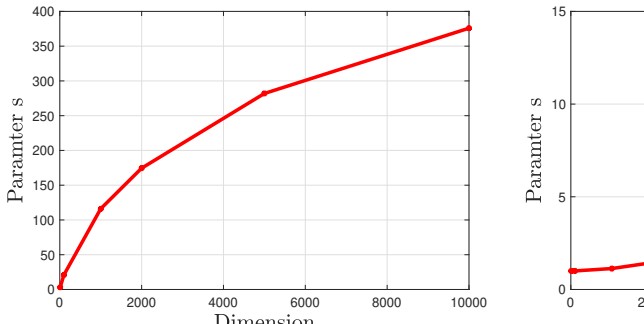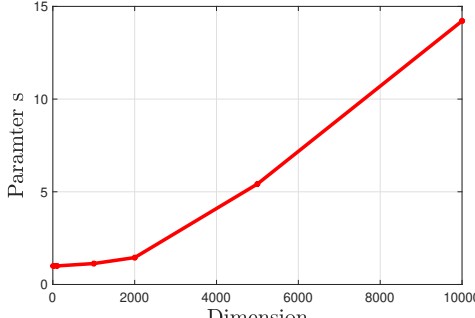

Figure 7: Parameter $s$ versus dimension $d$. The left plot corresponds to the case that all features have the same variance and the right plot corresponds to the case that the variance of one feature is much larger than the rest of the features.

Table 3: Parameter $s$ for different datasets.

| Dataset | MNIST | GISETTE | Orange | Epsilon |
|---|---|---|---|---|
| Dimension $d$ | 784 | 5000 | 14000 | 2000 |
| Parameter $s$ | 1.6 | 4.43 | 6.49 | 2.56 |

$\frac{1}{n}\sum_{k=1}^{n}\mathbf{x}^{(k)}\mathbf{x}^{(k)\top}$. We consider two cases. In the first case, we generate all the features according to the distribution $x_i \sim \mathcal{N}(0,1)$, and thus the expected risk Hessian is $\nabla^2 R(\mathbf{w}) = I$. In the second case, we generate the first coordinate according to $x_1 \sim \mathcal{N}(0,100)$ and the rest of the coordinates are generated based on $x_i \sim \mathcal{N}(0,1)$. In this case, we have $\nabla^2 R(\mathbf{w}) = D$ where $D \in \mathbb{R}^{d\times d}$ is a diagonal matrix with $D(1,1) = 100$ and $D(i,i) = 1$ for $2 \le i \le d$.

We plot the results in Figure 7 to present how parameter $s$ behaves as the dimension $d$ grows for these two scenarios. As shown in the right plot of the Figure 7, the parameter $s$ always stays very small, in the second scenario, no matter how large the dimension $d$ is. This case corresponds to the second scenario where the variance of one of the features is much larger than the rest of the features and therefore, the Frobenius norm and operator norm of the difference matrix $\nabla^2 R_n(\mathbf{w}) - \nabla^2 R(\mathbf{w})$ are very close to each other. So in the best case, $s$ is almost independent of the scale of dimension $d$. Note that in this case for $d = 10,000$, the parameter $s$ is 14, which shows that $s = O(1)$.

The left plot of Figure 7 shows that even in the worst case, where all features have similar variances/variations (the first scenario), the parameter $s$ could still be much smaller than the problem dimension $d$. Indeed, in this case, $s$ is larger than the previous studied case. However, it is still much smaller than dimension $d$ and it does not grow linearly with the dimension. In fact, for $d = 10,000$, the value of $s$ is $s = 380$ which is much smaller than $d$.

The above toy example showed that often parameter $s$ is much smaller than the problem dimension $d$. Next, we try to numerically identify the value of $s$ for the four datasets that we considered in our numerical experiments. Note that, in this case, we are not able to compute the Hessian of expected risk exactly as the underlying probability distribution is unknown. Instead, we approximate that by the Hessian of an empirical risk that is defined by a very large number of samples. Specifically, for each dataset, we use the Hessian of the ERM problem computed by all available samples $\nabla^2 R_N(\mathbf{w})$ in the training set as an approximation of the expected risk Hessian $\nabla^2 R(\mathbf{w})$. Hence, we compute the parameter $s$ by looking at the ratio of the Frobenius norm and operator norm of the difference matrix $\nabla^2 R_{m_0}(\mathbf{w}) - \nabla^2 R_N(\mathbf{w})$. The results are summarized in Table 3. As we observe, the parameter $s$ for all datasets stays close to 1 and is not within the same order as the problem dimension $d$. In fact, for all considered datasets, $s$ is less than 10, while dimensions are much larger.

Leveraging the results from numerical experiments in Figure 7 and Table 3, we argue that in most cases we are in the regime that $s = \mathcal{O}(1)$ and is independent of dimension $d$. Now recall that our lower bound on $m_0$ is $\Omega\left(\max\left\{d, \kappa^2 s \log d\right\}\right)$. These observations imply that, the worst theoretical lower bound of $\Omega\left(\kappa^2 d \log d\right)$ (where $s = O(d)$) rarely holds in practice, and in most common cases, we can assume that the lower bound on the initial sample size $m_0$ is $\Omega\left(\max\left\{d, \kappa^2 \log d\right\}\right)$.