# OpenReview forum: "Exploiting Local Convergence of Quasi-Newton Methods Globally: Adaptive Sample Size Approach"
_NeurIPS.cc/2021/Conference — NeurIPS 2021 Poster_

### Official Review · Reviewer_4SN8 · 2021-07-07

**Rating:** 7
**Confidence:** 5

**Summary:**

This paper
- introduces a new quasi-Newton (QN) algorithm called AdaQN to solve the ERM problem. This method is adaptive since it increases the sample size after few (3) runs of BFGS on a small ERM subproblem.
- gives its corresponding convergence analysis using a Matrix Bernstein bound
- explains that AdaQN perfoms well because of the superlinear convergence on each subproblem which is possible when $m_0$ is large enough (greater than $\mathcal{O} (\kappa^2 log (d))$ if we leave out $s$)
- provides clear comparison in term of theoretical computational cost is made against AdaNewton in Table 1
- performs numerical experiments showing the efficiency of the methods on small and medium size problems for l2 regularized logistic regression

**Limitations And Societal Impact:**

The authors did not really talk about the case where $m_0$ is in fact large.
No real experiments for very ill-conditioned problem.


**Main Review:**

The paper is really clear.
I really liked the idea of exploiting this local superlinear convergence of the QN methods successively, by having $S_1 \subset S_2 \subset ...$. The paper explains carefully its concepts, its algorithm implementation compares its cost against AdaNewton and then provides an easy to follow convergence analysis.

My first main concern (that I will develop below) is that the condition on $m_0$ that is $m_0 = \Omega (\kappa^2 s\log d)$ is to my opinion completely undermined by the authors.
My second main concern is the experimental part wish I find unclear and unfair. I develop this below.


Here are some comments, critics and questions:

Style/typo comments:
- 1) lines 37 and 74/75: not consistent writing of Quasi-Newton / quasi-Newton
- 2) line 48, not nice looking line because of \emph
- 3) line 77: missing word before reference [20]
- 4) line 106: ERM already defined earlier
- 5) line 111: $w^*$ is an optimal solution of the expected risk minimization. The authors will agree that $R$ is a function not an optim problem
- 6) line 264: typo, remove "the"
- 6 bis) line 552-553: strangely few lines are not numbered + typo " inequality(notice)" also there at line 570

Minor comments:
- 7) line 77: What did you mean when saying "To exploit quasi-newton methods for the described adaptive sample scheme, however ..." ? To be corrected I think
- 8) line 7 of Algo. 1: a break-line for $y$ would make it way easier to read
- 9) $\mu$ is not really standard for a regularization parameter, I would prefer $\lambda$ then say $\mu \approx \lambda$
- 9 bis) line 373: cite maybe the most recent version of Bottou and Bousquet 2011

Major comments:
- 10) A long comparison is made against AdaNewton, yet very few comparison is made agaisnt other (stochastic) QN methods like L-BFGS, which is known to perform very well in a lost of case, or the more recent K-BFGS ("Practical quasi-newton methods for training deep neural networks", Goldfarb, Ren, Bahamou, 2020) or Stochastic Block BFGS (Stochastic Block BFGS: Squeezing More Curvature out of Data, Gower, Goldfarb, Richtarik, 2016)
- 11) My main concern (that I will develop below) is that the condition on $m_0$ that is $m_0 = \Omega (\kappa^2 s\log d)$ is to my opinion completely undermined by the authors.  I would have liked a clear explanation about what happens when we are facing a very ill-conditioned problem such that $\kappa = N$ or even worse  $\kappa >> N$. What does AdaQN become in such a case ? Does it become QN with a constant sample size $m\equiv N$ ? In this case isn't it way more costly than 1st order methods ? What about numerical experiments in this case ?
- 12) In Prop. 1 and elsewhere in the paper, you clearly explain that having $S_1 \subset S_2 \subset ...$ is necessary for AdaQN to work. I have been through the proof and I was expecting that you would have to deal with a problem : at each outer loop iteration the points are \textbf{not} sampled independently. How is that it does not appear in your proof ?
- 13) Remark 4: Don't you omit the matrix inversions and the matrix vector product required by AdaQN when comparing it to Katyusha ? In other words, is this comparison of gradient complexity really fair between a first and a second order method ?
- 14) How can you get a condition number which equals 1 in your experiments ? Normalizing can improve the condition number, but I have never seen such a bound related to just simple normalization.
- 15) It is not clear what is monitored by "time" in your experiments. So I have been into the code and I noticed that the initialization which requires inverting the first hessian is not taken into account... In addition, SAGA is not initialized with a full gradient step, plus SAGA algorithm was launched with a decreasing step size which is unnecessary. This is one of the main advantage of VR methods. This is wat I really doubt about the fairness of the experiments.
- 16) In all your experiments $N$ is at most of medium size, you should have tried on RCV1.binary for instance $N = 600K$ to see if first order methods are still performing poorly compared to AdaQN.
- 17) Regularization parameters in table 2. look way to large... It is not representing the range of lambda used in practice. I would recommend $10^{-4}$ or $10^{-6}$. Moreover we could maybe see this way an experiment with large $\kappa$


Questions:
- 18) Why is AdaQN adaptive since you basically double the sample size at each iteration, so it's not dependent on anything else than the current number of samples m ?
- 19) Why didn't you include L-BFGS in your experiments ?
- 20) With is your initial point set to 1 in your experiments ?

---------------------------------------------------------------------------------------------------------------

**Update after rebuttal and exchanges with the authors and the other reviewers**

Main concerns I had about experimental settings, I think have been addressed. I think the authors must detail the experimental settings in the appendix such that the community can understand their numerical experiments and really be convinced that the settings are fair.

Despite the discussion on the practical "leveraging of superlinearity", I finally decide to **increase my score** but only up **to 6**.


---------------------------------------------------------------------------------------------------------------

**Final words**

In his updated review, reiewer RQBF wrote :

>   [...] after a discussion with the authors I concluded that the BFGS Hessian updates have basically no effect on the complexity analysis.

I agree with him on this point (and I think the authors too). Yet on my side, this discussion does not diminish the ideas nor the quality of the paper. Moreover, as I said above, my concerns about numerical experiments will be addressed in the final version of the paper (with fairer comparisons). My final thoughts about the paper are that despite this concern about superlinarity, it can be very profitable for the QN community and optimizers interested in adaptive mini-batch/sample size.
Hence, I increase my score to 7 and argue in favor of accepting this paper with one condition : a clear and important remark about this discussion on superlinearity must be added and highlighted.

**Time Spent Reviewing:**

9

---

> ### Author Response · Authors · 2021-08-10
> **Response to Reviewer 4SN8**
>
> We thank the reviewer for their feedback. The minor issues pointed out by the reviewer (typos, minor writing mistakes, ...) will be addressed in the revised paper. Next, we address the major comments raised by the reviewer.
>
> $~$
>
> **Comment 10:** Lack of comparison with other BFGS-like methods.
>
> **Response:** In our submission, we numerically compared AdaQN with a few BFGS-type methods, such as the original BFGS method and the stochastic quasi-Newton (SQN) method proposed in (A Stochastic quasi-Newton Method for Large-scale Optimization, Byrd et al. 2016). We thank the reviewer for mentioning (Practical quasi-Newton Methods for Training Deep Neural Networks, 2020) and (Stochastic Block BFGS: Squeezing More Curvature out of Data, 2016). We'll include these papers in the literature review of our paper and compare them with AdaQN numerically in the revised paper. Following your suggestion, we compared AdaQN with L-BFGS and summarize our results in the following paragraphs. Our experiments showcase the advantage of AdaQN with respect to L-BFGS.
>
>  For the MNIST data set, to reach the training error of $10^{-4}$ each method requires the following number of passes over the dataset:
>
>  AdaQN: 1.0 $\quad$ AdaNewton: 0.8 $\quad$ BFGS: 5.5 $\quad$ L-BFGS: 5
>
>  and the following convergence time (sec):
>
>  AdaQN: 0.3 $\quad$ AdaNewton: 0.2 $\quad$ BFGS: 2.0 $\quad$ L-BFGS: 1.3
>
>  For the MNIST data set, to reach the test accuracy of $97$% each method requires the following number of passes over the dataset:
>
>  AdaQN: 0.4 $\quad$ AdaNewton: 0.3 $\quad$ BFGS: 4.0 $\quad$ L-BFGS: 4.0
>
>  and the following convergence time (sec):
>
>  AdaQN: 0.2 $\quad$ AdaNewton: 0.1 $\quad$ BFGS: 0.7 $\quad$ L-BFGS: 0.6
>
> $~$
>
> **Comment 11:** My main concern is that the condition on $m_0$ is to my opinion completely undermined by the authors.  I'd have liked a clear explanation about what happens when we are facing a very ill-conditioned problem $\kappa \gg N$.
>
> **Response:** We'd like to mention that in several places in the paper we emphasize that our theoretical guarantees only hold for the case that $m_0$ is sufficiently large, and, hence, our approach is beneficial when the total number of samples $N$ is much larger than the lower bound on $m_0$.
>
> For instance, in the abstract of the paper (line 13-14) we state that "*We show that if the initial sample size is sufficiently large ....*", and in the  introduction of the paper (lines 91-94) it reads "*In our main theorem we show that $m_0$ is lower bounded by $\Omega(\max(d, \kappa^2s\log{d}))$, where .... . Hence, in this paper, we focus on the regime that the total number of samples satisfies $N \gg \max(d, \kappa^2s\log{d})$*". Moreover, in our main theorem (Theorem 1, Equation (9)) we explicitly mention that $m_0$ should satisfy the condition   $ m_0 = \Omega(\max  (d, \kappa^2s\log{d} ))$. We even
>  admitted this condition as one of the weaknesses of our paper in Section 6 (line 362 to 364), as we stated that "*Our result shows AdaQN provably outperforms other benchmarks only in the regime that $N \gg  \max(d, \kappa^2s\log{d})$. This is due to the lower bound on the size of initial training set $m_0$.*".
>
>  As discussed above, the proposed method is beneficial when we are in the regime that $N \gg  \max(d, \kappa^2\log{d})$, and if this condition does not hold, for instance $\kappa \gg N$, then the size of the initial training set is larger than the total number of samples that we have access to and the proposed AdaQN becomes vacuous. We'd like to reiterate that we only claim a meaningful gain for AdaQN when we are in the regime of $N \gg  \max(d, \kappa^2\log{d})$. Please note that in many large-scale learning problems where $N$ is extremely large this condition holds. Although this point has mentioned in the original submission, we'll highlight it as a remark in the revised paper to address your concern.
>
> $~$
>
> **Comment 12:** Questions about the sample data dependency.
>
> **Response:** As mentioned in lines 114-155, sample points are drawn independently in our algorithm. Basically, when we aim to move from the setting with $m$ samples to the setting with $n=2m$ samples, we randomly pick $m$ new samples from the sample points that are not used so far and add them to the $m$ samples that we have used in the previous stage. This way, the samples are still i.i.d. and there is no correlation between the selected samples as has been shown in prior work on adaptive sample size methods, including Ada Newton in [19] and the BET algorithm mentioned by Reviewer RQFB.
>
> More precisely, it will not lead to any issue as we assume that "*all the expectations are with respect to the corresponding training set*"; see lines 126-127. Therefore, for the training sets $S_m, S_n$ with $m < n$, where $ S_m\subset S_n$, and their corresponding objective functions $R_m(\mathbf{w})$ and $R_n(\mathbf{w})$, we have
>
> $$
> E\_{S\_n}[R\_n(\mathbf{w})] = R(\mathbf{w}),
> $$
>
> $$
> E\_{S\_n}[R\_m(\mathbf{w})] = E\_{S\_{n - m}}[E\_{S\_m}[R\_m(\mathbf{w})]] = E\_{S\_m}[R\_m(\mathbf{w})] = R(\mathbf{w}),
> $$
>
> where $R(\mathbf{w})$ is the expected loss function. In the revised paper, we will highlight this point in the proof of Proposition 1.
>
> $~$
>
> **Comment 13:** Remark 4: Don't you omit the matrix inversions and the matrix vector product required by AdaQN when comparing it to Katyusha?
>
> **Response:** This is a valid point. In our main theorem, when we report the overall complexity of the AdaQN method we mention that it requires computing one matrix inversion, $m_0$ Hessian evaluations, and  $3\log(N/m_0)$ matrix-vector products. However, in Remark 4, when we compare AdaQN with Katyusha, we only compare their gradient complexity. The reviewer point is indeed valid, and in the revised paper, we will emphasize that the gradient complexity of AdaQN is better than Katyusha, while it requires evaluating an additional matrix inversion and some Hessian and matrix-vector product computations. It is worth noting that in many cases, the cost of computing a matrix-vector product is the same order as the cost of a single gradient computation.
>
> $~$
>
> **Comment 14:** Problems about the condition number in experiments.
>
> **Response:** We'd like to clarify that the problem condition number in our experiments is not $1$. In our experiments, we normalize the data points and as a result the smoothness parameter for all our problems becomes $L=\mu+1$, where $\mu$ comes from the regularization term and $1$ comes from the regression term with normalized data points. As a result, the condition number of the problem is $\kappa = \frac{L}{\mu} = 1+\frac{1}{\mu}$. Hence, roughly speaking, the condition number scales as $\kappa \approx \frac{1}{\mu}$, where $\mu$ is the regularization parameter and the strong convexity constant. Note that in our experiments, the parameter $\mu$ is often small, and thus, the condition number is large.
>
> $~$
>
> **Comment 15:** Unfair and unclear in numerical experiments.
>
> **Response:** The reviewer is right about the problems in our numerical experiments. Following your suggestion, in our revised experiments, we incorporated the computation time of the first (and only) Hessian inversion in AdaQN. Moreover, we initialized the SAGA algorithm with a full gradient and hand-tuned a fixed step size for each dataset to obtain the best performance. After these modifications, we observed that the performance of SAGA improves and the convergence time of AdaQN becomes slightly worse, but their relative performance does not change significantly. This is mainly due to the fact that AdaQN requires only one matrix inversion. Next, we report the results of our new experiments on four datasets:
>
> Convergence time (sec) required to reach the test accuracy of $97$% for the MNIST data set:
>
> AdaQN: 0.2 $\quad$ AdaNewton: 0.1 $\quad$ SAGA: 1.0
>
> Convergence time (sec) required to reach the test accuracy of $90$% for the GISETTE data set:
>
> AdaQN: 1.8 $\quad$ AdaNewton: 2.5 $\quad$ SAGA: 4.0
>
> $~$
>
> **Comment 16:** The total sample size $N$ is small.
>
> **Response:** The largest dataset that we considered in our experiments is the Epsilon dataset with $80K$ samples, which is a relatively large dataset. But as the reviewer suggested, it would be useful to study the behavior of AdaQN on  larger datasets, such as RCV1 with $600K$ samples. We'll conduct such experiments in the revised paper.
>
> $~$
>
> **Comment 17:** Regularization parameters are small.
>
> **Response:** For our experiments, we tried different choices of regularization parameters for different datasets, and finally reported the one that led to best test error. In fact, selecting a smaller regularization parameter would increase the condition number of the problem and make AdaQN more favorable with respect to first-order methods. For instance, in the last experiment on Epsilon dataset where $\mu$ is very small ($\mu=10^{-4}$), the advantage of AdaQN is more significant.
>
> $~$
>
> **Comment 18:** The meaning of "adaptive".
>
> **Response:** The reason that we call our method "adaptive" is because the "active sample size" changes over time and we operate on different ERM problems throughout the training process. For this reason, as it has been named previously in the literature, we call it an "adaptive sample size method".
>
> $~$
>
> **Comment 19** Experiments of L-BFGS.
>
> **Response:** Thanks for your suggestion. Please check  our answer to Comment 10 to see the results for the L-BFGS method.
>
> $~$
>
> **Comment 20:** Problems about the initial point.
>
> **Response:** Note that in our experiments, to find an initial point that is within the statistical accuracy of the ERM problem with $m_0$ samples, we run multiple updates of gradient descent starting from the point $[1,\dots,1]$. The output of this procedure is considered as the initial stage for all the algorithms. Hence, the choice of the first point $[1,\dots,1]$ does not influence our empirical results.  We will highlight this point in the revised paper.

---

> > ### Comment · Reviewer_4SN8 · 2021-08-19
> > **New comments**
> >
> > I thank the authors for the clarity of their explanations.
> >
> > First, I'm glad that the authors ran experiments vs L-BFGS and that my comments lead to fairer comparisons with other algorithms like Katyusha and in the numerical experiments with the other methods by taking into account preprocessing time.
> >
> > I agree with the fact that the paper focus on cases where the optimization problem is such that $\kappa < N$ and that this limitation is already described in the paper. Yet, I would have been curious to see how it behaves in such very ill-conditioned problem, but this is a side thought.
> >
> > Also, I now agree and understand points in comments 11, 12, 14 and 17.
> >
> > I still have 1 or 2 questions/comments :
> > - **Comment 18** : I see your point. Even if the community refers to this kind of procedure I feel it is a bit misleading since in BET or in AdaQN one basically always increases the batch size using the same procedure no matter what are the local information available. In my opinion, a real "adaptive" sample size would be related to local information as the Polyak step size is adaptive since it is based on the value function and the gradient at the current iterate. But once again, this is a discussion out of the scope of the paper.
> >
> > - **Comment 20** : You answered that
> > > we run multiple updates of gradient descent starting from the point $[1,...,1]$
> >
> > Is this done only for AdaQN ? If yes, this "warm up" should be taken into account as passes over the data / initialization time. But I don't think first order methods would benefit that much of this initialization as opposed to QN methods.
> > In my opinion, the fairest way to compare AdaQN (or other QN methods) vs 1st order methods like SAGA, SGD etc in your experiments would be to compare :
> > - Start from 1, initialization with GD steps, then AdaQN
> > -  Start from 1, SAGA or SGD or KAtyusha
> >
> >
> > I am open to reconsider my score but I wait to see what the other reviewers have to say, especially reviwer RQFB about this debate whether AdaQN * “exploits the superlinear convergence”* or not.

---

> > > ### Author Response · Authors · 2021-08-20
> > > **Response to New Comments of Reviewer 4SN8**
> > >
> > > We thank the reviewer for reading our response. In the following paragraphs, we address the new comments raised by the reviewer.
> > >
> > > $~$
> > >
> > > **How does your algorithm behave in settings where $\kappa$ is much larger than $N$?**
> > >
> > > This is a good point. Based on our theoretical bounds, when the condition number $\kappa$ is very large, and we are in the regime that $\kappa > N$, the lower bound for the initial sample size $m_0$ becomes larger than $N$. In this case, our adaptive sample size quasi-Newton method is equivalent to the classical BFGS algorithm, as we need to use all the available samples for the first problem. Hence, our theoretical bounds do not improve AdaQN in this setting compared to the classical quasi-Newton methods.
> > >
> > > However, in practice, AdaQn still works pretty well in very ill-conditioned problems ($\kappa>N$), when we use a small initial sample size $m_0$, that may not satisfy our theoretical bounds. For example, in our numerical experiments on the Epsilon dataset, the strong convexity parameter is $\mu = 10^{-4}$, and as a result, the condition number is  $\kappa\approx 10^4$. In this case, we set the initial sample size to be $m_0=4096$, which is much smaller than the required condition by our theoretical results. Nevertheless, AdaQN still works pretty well in this experiment and outperforms the classic BFGS method and other considered algorithms. As mentioned previously, we believe that the main reason for this mismatch is that the recent non-asymptotic convergence analysis of the BFGS method that we exploit in our analysis may not be tight, and there is room for improving this bound. Specifically, establishing a less strict condition on the initial Hessian approximation error of BFGS required for achieving a superlinear convergence rate could lead to a much smaller lower bound for the parameter $m_0$.
> > >
> > > $~$
> > >
> > > **Comment about the use of “adaptive” for our algorithm**
> > >
> > > This is a valid point by the reviewer. We agree with the reviewer that calling our approach ``adaptive sample size" might be a bit confusing. In the revised paper, we will include a remark and explicitly mention that we basically mean an increasing batch-size scheme by adaptive sample size. Thanks for raising this point.
> > >
> > > $~$
> > >
> > > **Comment about the initialization scheme**
> > >
> > > The warm-up stage of running multiple updates of gradient descent starting from the vector $[1,\dots,1]$ is not for the adaptive quasi-Newton method only. In fact, we applied this warm-up procedure for all the algorithms we used in our experiments. To be more specific, we start from $[1,\dots,1]$ and then run gradient descent updates to find a vector $w_{init}$ that is within the statistical accuracy of ERM with $m_0$ samples. Then, we use $w_{init}$ as the initial iterate for all algorithms to have a fair comparison.
> > >
> > > The reviewer's suggestion for the initialization scheme is also valid, and in the final version of the paper, we will also use the initialization approach suggested by the reviewer.
> > > By following the reviewer's suggestion, we should also incorporate the cost of solving the first ERM problem with $m_0$ samples for AdaQN in our comparison, while the remaining methods do not require such computation as they can start at $[1,..,1]$. On the other hand, we should add that the cost of running gradient descent for solving the ERM problem with $m_0$ samples is negligible in all our experiments for AdaQN, as this cost scales with $m_0$, and in all our experiments $m_0\ll N$.
> > >
> > > Following your suggestion, We conduct new numerical experiments on the MNIST dataset using your suggested initialization. As described above, in this setup, for AdaQN, we run a warm-up step to find $w_{init}$, while for first-order methods, we simply start from $[1,\dots,1]$. In this case, we also take into account the cost of the warm-up stage for AdaQN.
> > >
> > > For the MNIST data set, to reach the training error of $10^{-2}$ each method requires the following number of passes over the dataset:
> > >
> > > AdaQN: 2 $\quad$ SGD: 18 $\quad$ SAGA: 13 $\quad$ Katyusha: 10
> > >
> > > and the following convergence time(second):
> > >
> > > AdaQN: 0.4 $\quad$ SGD: 12 $\quad$ SAGA: 6 $\quad$ Katyusha: 3.5
> > >
> > > For the MNIST data set, to reach the test accuracy of $97\\%$ each method requires the following number of passes over the dataset:
> > >
> > > AdaQN: 2 $\quad$ SGD: 11 $\quad$ SAGA: 4 $\quad$ Katyusha: 7
> > >
> > >  and the following convergence time (seconds):
> > >
> > > AdaQN: 0.3 $\quad$ SGD: 2.5 $\quad$ SAGA: 2 $\quad$ Katyusha: 0.6
> > >
> > > $~$
> > >
> > >  **Comment about the fact that AdaQN “exploits the superlinear convergence” or not**.
> > >
> > > In our response to the comment by Reviewer RQFB, we clarified why and how we use the superlinear convergence rate of $\mathcal{O}((1/k)^k)$ quasi-Newton methods in our analysis. Here, we want to emphasize that the key step that exploits this property of quasi-Newton methods is in the proof of Theorem 1 (Appendix B). Specifically, consider $t_n$ as the number of iterations required to solve the ERM problem with $n$ samples. When we try to establish an upper bound on $t_n$, we use the fact that after $t_n$ iterations, the error is bounded above by $(1/t_n)^{t_n} e_0$, where $e_0$ is the initial error. Indeed, this error bound comes from the fact that BFGS converges at a superlinear rate of $(1/k)^k$. Using this bound, we then show that if $t_n$ is $3$, the error has decreased sufficiently, and we have reached the statistical accuracy of the problem with $n$ samples. For more details, please check our response to Reviewer RQFB.

---

> > > > ### Comment · Reviewer_rmWp · 2021-08-20
> > > > **Thanks for the authors' responses**
> > > >
> > > > I have read the authors' responses. It does not change my opinion that this paper is on the borderline. I also agree with the other reviewers' comments on the unclarity of the paper, especially some confusing numerical settings, and the mismatching between the current theory and the good experiments.
> > > >
> > > > As the authors also agree that there is room for improving their bound, I also expect better theoretical results in the topic of quasi-Newton methods.
> > > >
> > > > Moreover, the authors mentioned their analysis built on the worst case, including hard instances or corner cases, which could not explain the numerical results they discovered.
> > > > I suggest the authors analyze simple objectives (such as the quadratic function), because simple cases may provide intuitive and promising findings and guide improvements.

---

### Official Review · Reviewer_rmWp · 2021-07-16

**Rating:** 6
**Confidence:** 3

**Summary:**

This paper introduces an adaptive sample size quasi-Newton (AdaQN) method that exploits the superlinear convergence of quasi-Newton methods globally and throughout the entire learning process. The authors also provide convergence analysis to verify the AdaQN method, showing less gradient computation, as well as computational cost compared with previous SOTA methods. The numerical experiments on various datasets also confirm their theoretical findings.


**Limitations And Societal Impact:**

The only concern is the mismatching of m0 in experiments and theory. At first glance from Theorem 1, I consider such an m0 is only suitable for the well-conditioned case that $N \geq \kappa^2\log d$. However, the numerical experiments show that the initial sample size m0 could be much smaller than the threshold.  Although the authors believe this limitation could be resolved by a tighter non-asymptotic analysis of the BFGS method, I could not believe the explanation from the large gap in theory and experiments. Thus I keep a conservative assessment.

Overall, I am willing to defend my assessment and am unfamiliar with some pieces of related work. Proof details were not carefully checked.

[1] Mokhtari, A., & Ribeiro, A. (2017, December). First-order adaptive sample size methods to reduce complexity of empirical risk minimization. In Proceedings of the 31st International Conference on Neural Information Processing Systems (pp. 2057-2065).


**Main Review:**


I consider this work gives an effective QN method through interesting intuition using adaptive sample size. Adopting adaptive sample size arises in previous works [1], but not for Quasi-Newton methods, and it indeed provides convincing results. Therefore, I still approve of the originality of this work. Moreover, the work is well-written and has high quality and clarity.

The exploitation of the superlinear convergence of quasi-Newton methods globally is an important problem in my view. As far as I know, previous results build on a small local neighborhood of the optimal solution as the authors mentioned. Therefore, the significance of this work is undoubted.



**Time Spent Reviewing:**

8

---

> ### Author Response · Authors · 2021-08-10
> **Response to Reviewer rmWp**
>
> We thank the reviewer for the constructive feedback. Next, we briefly address the main issue raised by the reviewer.
>
> **Comment:** The mismatching of $m_0$ in experiments and the theory.
>
> **Response:** As the reviewer has correctly pointed out,  in our current theory, we  require the size of the initial training set $m\_0$ to satisfy the condition $m\_0 \geq \kappa^2\log{d}$. As a result, we need the total number of samples $N$ to be larger than this lower bound to obtain a meaningful gain for AdaQN. On the other hand, our numerical experiments show that the initial sample size $m\_0$ could be much smaller than the above threshold. As we mentioned in our submission, we believe that the main reason for this mismatch is the fact that the recent non-asymptotic convergence analysis of the BFGS method that we exploit in our analysis may not be tight, and there is room for improving this bound. Specifically, establishing a less strict condition on the initial Hessian approximation error of BFGS required for achieving a superlinear convergence rate could lead to a much smaller lower bound for the parameter $m\_0$.
>
> Another possible reason for the gap between our theory and experiments is that our analysis is the worst-case analysis and it studies a lower bound for $m\_0$ that works for any setting. In other words, there could be some hard instances or corner cases that our current experiments do not capture, and for these hard instances we might need the number of samples required by our theoretical study. Identifying hard instances of empirical risk minimization problems for which a larger initial sample size is required is a future research direction that we plan to investigate.

---

### Official Review · Reviewer_RQFB · 2021-07-17

**Rating:** 5
**Confidence:** 5

**Summary:**

The paper proposes an optimization method for empirical risk minimization, combining two techniques from the literature: (1) adaptive sample sizes, where the optimization algorithm is initially ran on a small batch of data, which is then gradually increased by doubling it every couple iterations; and (2) quasi-Newton methods for updating the iterates, such as the BFGS algorithm. The authors leverage recent results on non-asymptotic local convergence analysis of BFGS and show that with an appropriate choice of sample size, the quasi-Newton method will stay in its local convergence regime throughout the optimization, leading to strong convergence guarantees.

**Limitations And Societal Impact:**

Yes.

**Main Review:**

Thank you to the authors for the response and the following discussion. The authors have clarified that superlinearity of BFGS is not needed to establish their results, and they promised to clarify this in the paper, which is good. They also promised to include in the future version of the paper a discussion and empirical results comparing to the prior work of [DMKVW18], which is appreciated. On the other hand, in my view a significant part of the technical novelty of the approach rested on the apparent reliance on the superlinear nature of the BFGS updates, but after a discussion with the authors I concluded that the BFGS Hessian updates have basically no effect on the complexity analysis. This weakens the paper in my view. Thus, all in all, I maintain my score.

First, there is one important related work that is missing from the literature review, namely [DMKVW18]. In fact, that paper is to my knowledge the first to propose using adaptive sample sizes with a quasi-Newton method (in their case, L-BFGS). However, it does not provide a convergence analysis that is specific to any particular quasi-Newton method, which is the main contribution of the present submission. Instead, they show that if for each sample size an optimizer exhibits a convergence rate of at least, say, (1/2)^t, then one should double the sample size at each stage after a constant number of iterations (same as in the present submission). Thus, in principle, one could combine that result with the recent work on local convergence rates of BFGS [21-23] to establish a guarantee analogous to the one obtained by the authors. Of course, this is by no means straightforward, but it should at least be addressed in the submission. I would also be interested to see an empirical comparison of AdaQN with the BET method proposed by [DMKVW18] (ran with L-BFGS), since that can be considered as the closest baseline from the literature.

Second, I take issue with the claim that the proposed approach “exploits the superlinear convergence” of quasi-Newton methods. As the authors discuss, the superlinear convergence of BFGS takes the form of roughly (1/t)^t, where t is the number of iterations performed in one stage of Algorithm 1. However, since the algorithm performs only 3 iterations per stage, then it has no opportunity to exploit superlinear convergence, and all you actually obtain is linear convergence that is independent of the condition number. In fact, the proposed analysis demonstrates that the superlinear convergence doesn’t really matter here, because we reach statistical accuracy before it kicks in. So, I think the authors should not say that the algorithm is “exploiting the superlinear convergence”, but it is ok to say that the algorithm is “exploiting local convergence”, because that is still very much accurate.

The above discussion about the lack of true superlinear convergence highlights another issue with the algorithm, namely, with the way that the Hessian estimate is being updated. Basically, what the authors propose is that every stage is initialized with the Hessian estimate computed from the smallest sample size (stage 1), then three iterations of BFGS slightly improve that estimate, and then it goes back to the original one. This seems strange. Why not continue with the improved estimate and keep updating it with the BFGS updates? The authors partially address this in Remark 1, pointing out that the initial Hessian estimate is good enough to ensure the local convergence phase of BFGS. But then what is the point of the three updates to the estimate? In fact, I’m quite convinced that the convergence guarantee from Theorem 1 can be shown for the method that just uses the initial Hessian estimate without every updating it. But then, this is no longer BFGS or even quasi-Newton. This is why, in my view, the paper is to a certain extent misrepresenting the nature of why the proposed method achieves the convergence it achieves.

Overall, the strategy of combining adaptive sample sizes with quasi-Newton methods is very interesting, and it is very nice to have a complete convergence analysis of an algorithm that follows that strategy. However, I do have some serious concerns regarding the novelty and presentation of this paper. I am open to reconsider the score if those concerns are addressed in the response.

[DMKVW18] M. Derezinski, D. Mahajan, S. S. Keerthi, S. V. N. Vishwanathan, and M. Weimer. "Batch-expansion training: an efficient optimization framework." In International Conference on Artificial Intelligence and Statistics, pp. 736-744, 2018.

**Time Spent Reviewing:**

6 hours

---

> ### Author Response · Authors · 2021-08-10
> **Response to Reviewer RQFB**
>
> We thank the reviewer for the constructive feedback. Next, we briefly address the issues raised by the reviewer.
>
> **Comment:** Comparison with the BET method proposed in [DMKVW18].
>
> **Response:**
> First, we thank the reviewer for bringing this paper to our attention. Indeed, this paper is related to our submission, and we will discuss it in detail in the revised version of our paper. We would like to add that the  Batch-expansion training (BET) method in [DMKVW18]  differs from our proposed framework in terms of algorithm development and convergence analysis. Specifically, the BET method uses a similar idea of geometrically increasing the size of the training set while using the L-BFGS method for solving the Empirical Risk Minimization (ERM) subproblems. The authors leverage the *linear* convergence rate of the L-BFGS method to characterize the overall complexity of the BET algorithm. Importantly, since this work focuses on a linearly convergent quasi-Newton algorithm, the overall complexity of BET may not be better than the one for adaptive sample size first-order methods studied in the following paper:
>
> Mokhtari, A. and Ribeiro, A. ``First-order adaptive sample size methods to reduce complexity of empirical risk minimization". NeurIPS 2017.
>
> We should emphasize that this is not a limitation of the analysis in [DMKVW18]. It is simply caused by the fact that the L-BFGS method converges linearly (even locally), similar to first-order methods. In fact, its linear convergence contraction factor is not provably better than the one for first-order methods.
>
> In our submission, we leverage the local *superlinear* convergence rate of the BFGS method to improve the overall complexity of adaptive sample size first-order methods, and it also improves the sample complexity of BET. Our analysis heavily builds on the fact that the solution of the ERM problem with $m$ samples is within the superlinear convergence neighborhood of the BFGS method for the next ERM problem with $n=2m$ samples. As a result, BFGS solves each subproblem at a *superlinear* rate. Since we need to reach the statistical accuracy of each subproblem and there is no need to solve the subproblems exactly, we show that at most 3 iterations of BFGS is sufficient for solving each subproblem.
>
> Another important point that we need to emphasize is that since the iterates of AdaQN are always in a local neighborhood in which BFGS converges superlinearly, we can always set the step size as $1$, while for BET this argument does not hold and proper selection of the step size (based on a the smoothness parameter or a line-search technique) is needed.
>
> In our revised paper, we will discuss the BET method in the literature review of the paper and will compare it with our proposed method numerically. Our experiments during the rebuttal period show that BET performs well against first order methods (such as SAGA), but it underperforms compared to the proposed adaptive quasi-Newton method. Again, this is mainly due to the fact that, in each stage, the BET algorithm converges linearly while solving each subproblem, while the proposed AdaQN method converges at a superlinear rate. In the following paragraph, we summarize our experimental results:
>
> For the MNIST data set, to reach the training error of $10^{-4}$ each method requires the following number of passes over the dataset:
>
> AdaQN: 1.0  $\quad$    AdaNewton: 0.8  $\quad$    BET: 2.5
>
> and the following convergence time(second):
>
> AdaQN: 0.3   $\quad$   AdaNewton: 0.2  $\quad$    BET: 1.0
>
> For the GISETTE data set, to reach the training error of $10^{-4}$ each method requires the following number of passes over the dataset:
>
> AdaQN: 2.0 $\quad$ AdaNewton: 1.8 $\quad$ BET: 5.0
>
> and the following convergence time(second):
>
> AdaQN: 2.5 $\quad$ AdaNewton: 4.5$\quad$ BET: 6.0
>
> $~$
>
> **Comment:** AdaQN does not exploit superlinear convergence of BFGS.
>
> **Response:** We would like to emphasize that our algorithm development and  convergence analysis indeed exploit the local superlinear convergence rate of the BFGS method. Proposition 3 and Proposition 4 show that if the size of the initial training set is sufficiently large, then the solution of the problem with $m$ samples is within the superlinear convergence neighborhood of the problem with $n=2m$ samples. We use these results and the superlinear convergence ($O(1/k)^k$) of BFGS in the proof of Theorem 1 to show that running 3 iterations of BFGS is sufficient to solve each subproblem within its statistical accuracy. This point is discussed in detail in the proof of Theorem 1 (Appendix B5), and we briefly review this argument here as well.
>
> Suppose we are at the stage with $n=2m$ samples and the initial iterate $\mathbf{w}\_m$ is within the statistical accuracy of $R\_m$. Further, suppose the size of initial set $m_0$ satisfies the required conditions, and as a result, $\mathbf{w}\_m$ is within the superlinear convergence neighborhood of BFGS for the ERM problem with $n=2m$ samples.
> If we define $\mathbf{w}\_n$ as the output of the process after running $t\_n$ updates of BFGS on $R\_n$ with step size $1$, then based on the superlinear convergence rate of $O((1/k)^k)$, discussed in Proposition 2, we have
>
> $$
> \mathbb{E}[R_n(\mathbf{w}_n) - R_n(\mathbf{w}^*_n)] \leq 1.1 \ \left(1/{t_n}\right)^{t_n}\mathbb{E}[R_n(\mathbf{w}_m) - R_n(\mathbf{w}^*_n)].
> $$
>
> Moreover, based on Proposition 1, we have
> $\mathbb{E}[R_n(\mathbf{w}_m) - R_n(\mathbf{w}^*_n)] \leq 3V_m$. Combining these two inequalities implies that
>
> $$
>     \mathbb{E}[R_n(\mathbf{w}_n) - R_n(\mathbf{w}^*_n)] \leq 3.3\ V_m \left(1/{t_n}\right)^{t_n}.
> $$
>
> Our goal is to ensure that the output iterate $\mathbf{w}_n$ reaches the statistical accuracy of $R_n$ and satisfies
> $\mathbb{E}[R_n(\mathbf{w}_n) - R_n(\mathbf{w}^*_n)] \leq V_n$. This condition is indeed satisfied if the above upper bound is smaller than $V_n$ and we have
>
> $$
>      3.3\  V_m \left(1/{t_n}\right)^{t_n} \leq V_n.
> $$
>
> As $V_n = \mathcal{O}({1}/{n})$ and $n = 2m$, the above condition is equivalent to
> $\left({1}/{t_n}\right)^{t_n} \leq ({1}/{6.6}) $. It can be verified that this condition holds if $t_n$ satisfies
>
> $$
> t_n \geq 1 + \ln{(6.6)}.
> $$
>
> This expression shows that, in the stage that the number of active samples is $n$, we need to run BFGS for at most $t_n =3$ iterations (since $3 \geq 1 + \ln(6.6))$. Note that this threshold is independent of $n$ or any other parameters.
>
> The above discussion shows that we do exploit the superlinear convergence rate of $O((1/k)^k)$  in our analysis to show that each subproblem can be solved using at most 3 iterations.
>
> $~$
>
> **Comment:** Why not continue with the improved Hessian estimate and keep updating it with the BFGS updates?
>
> **Response:** As the reviewer has correctly pointed out, one reasonable approach for initializing the Hessian inverse approximation for each phase is to use the last Hessian approximation matrix from the previous stage, instead of using the Hessian matrix of the initial set with $m_0$ samples. This is indeed a natural approach to consider, and we did investigate this direction previously, but for two reasons we did not use this idea and decided to go with the same initial Hessian approximation at each stage instead.
>
> First, we observed in our numerical experiments that using the Hessian approximation from the previous stage as the initial Hessian approximation for the next stage behaves similarly to our proposed AdaQN which uses $H\_{m\_0}^{-1}$ as the initial Hessian inverse approximation for each stage.
>
> Second, we realized that theoretically we can not guarantee the last Hessian approximation from the phase with $m$ samples satisfies the conditions that we need for superlinear convergence of BFGS for the next subproblem with $n=2m$ samples. To be more specific, consider the case that we have $m_0$ samples in our active training set and we start our updates with the matrix $H\_{m\_0}^{-1}$. At the end of this process, we will end up with a new Hessian inverse approximation matrix which we here refer to as $\tilde{H}\_{m\_0}^{-1}$. In BFGS analysis, there is no guarantee that if we follow the updates of BFGS the Hessian approximation improves and as a result we cannot ensure that $\tilde{H}\_{m\_0}^{-1}$ is closer to the Hessian at the optimal point of the problem with $m_0$ samples compared to $H\_{m\_0}^{-1}$. Note that in the analysis of BFGS-type methods, we can only show that the descent direction improves, and it cannot be shown that the Hessian approximation improves as time progresses. Considering these points, we cannot show that if $H\_{m\_0}^{-1}$ satisfies the required condition for the initial Hessian approximation of BFGS the same holds for updated approximation matrix $\tilde{H}\_{m\_0}^{-1}$.
>
> In our analysis, we ensure that $m_0$ is sufficiently large so that $H\_{m\_0}$ satisfies the required initial Hessian approximation necessary for superlinear convergence of BFGS for ERM with $n$ samples for any $n>m_0$, i.e., $||H\_{m\_0} - \nabla^2{R\_n(\mathbf{w}\_n^\ast)}||\_F \leq \delta$. As discussed above, we cannot ensure that after a few updates on the Hessian approximation this bound improves and the updated Hessian approximation, denoted by $B_n$, will satisfy $||B\_n - \nabla^2{R\_n(\mathbf{w}\_n^\ast)}||\_F \leq \delta$ or even the required condition for the next problem, which is $||B\_n - \nabla^2{R\_{2n}(\mathbf{w}\_{2n}^\ast)}||\_F \leq \delta$. This is simply due to the fact that the error of Hessian approximation in BFGS does not provably decrease. We will highlight this point in the revised paper.

---

> > ### Comment · Reviewer_RQFB · 2021-08-22
> > **Clarifying about superlinear convergence**
> >
> > I thank the authors for the detailed response as well as the additional empirical evaluation. Overall, I found the response helpful, however I still take issue with the claim that the paper is leveraging “superlinear convergence”. I think this is an important point that needs clarification in the paper, whereas based on the response, the authors disagree with me about that.
> >
> > To me, the difference between “superlinear” and “linear” convergence is precisely the difference between having $(1/t_n)^{t_n}$ in Proposition 2, and having, say, $(1/2)^{t_n}$, where $1/2$ is the rate of linear convergence (could be replaced by some other constant). In fact, I am fairly convinced that one could achieve this linear rate for Algorithm 1 (maybe up to absolute constants) even if we dropped all of the BFGS steps that update the Hessian approximation (i.e., lines 7 and 8). Now, my point is that even if we replaced Proposition 2 with this weaker convergence guarantee that does not in any way leverage the BFGS steps or "superlinearity", the rest of the analysis in the whole paper would still go through. In fact, as discussed  in the response, the only way that the convergence rate is used in the analysis is to ensure that $(1/t_n)^{t_n}\leq 1/6.6$, which requires $t_n=3$. If we replace this by, say, the condition that $(1/2)^{t_n}\leq1/6.6$, then we observe that $t_n=3$ also satisfies that guarantee, so all of the complexity analysis yields identical results. Of course, this depends on the constant $1/2$ in the rate, but as long as this is an absolute constant, we will get $t_n=O(1)$.
> >
> > So, could the authors clarify the following additional points:
> > 1. Is it true that the rest of the analysis in the paper would still go through if the convergence guarantee in Proposition 2 had $(1/2)^{t_n}$ (linear) instead of $(1/t_n)^{t_n}$ (superlinear)?
> > 2. Do you agree that this change makes the difference between "linear" and "superlinear" convergence?
> > 3. Do you believe it is true that Algorithm 1 without any Hessian updates (lines 7 and 8) achieves the convergence guarantee $(1/c)^{t_n}$ for some absolute constant $c>1$ (e.g., $c=2$) under the assumptions of Proposition 2?

---

> > > ### Author Response · Authors · 2021-08-23
> > > **Feedback to Reviewer RQFB**
> > >
> > > We thank the reviewer for reading our response. In the following paragraphs, we address the new comments raised by the reviewer.
> > >
> > > First, we briefly comment on the superlinear and linear discussion raised by the reviewer. To clarify this conversation, we think two points should be distinguished from each other. The first point is whether we use the superlinear convergence of BFGS in our analysis or not. And, the second point is whether one can achieve a similar guarantee, possibly with another algorithm that only converges linearly.
> > >
> > > Regarding the first point, we hope that the reviewer agrees that we exploit the superlinear convergence analysis of BFGS to establish our results, as explained in detail in our previous response. Regarding the second point, we agree with the reviewer that it might be possible to achieve a similar overall sample complexity if one uses a second-order method that locally converges linearly with a contraction factor that is independent of problem parameters, e.g., $1/2^k$. But, the required local conditions for that algorithm would be different from ours, and it would require establishing a novel analysis to find a lower bound on $m_0$. We explain this matter in the following paragraphs.
> > >
> > > To establish our result, we require a superlinear convergence rate for BFGS. This rate requires two conditions on the initial iterate error and the initial Hessian approximation error, as described in Proposition 2. As we discuss in the paper, to ensure that these conditions hold at the very first phase, we need to ensure that the size of the initial training set $m_0$ is larger than a specific threshold, where the conditions in Proposition 2 identify this threshold. Hence, this threshold is only valid for the case that we analyze the BFGS method.
> > >
> > > As the reviewer has pointed out, there could be other algorithms (possibly second-order) for which we can obtain a local linear rate of $1/2^k$. Indeed, using such a convergence rate, one could also show that a constant number of iterations is sufficient to solve each subproblem. However, to do so, we first need to understand what local conditions are needed for that algorithm to achieve such a rate and ensure that those local conditions for that method are satisfied. In other words, the questions that need to be answered are: What method and under what local conditions achieve a local convergence rate of $1/2^k$? Once these questions are answered, our analysis would be extendable to that algorithm, but it will lead to a different bound on $m_0$.
> > >
> > > Specifically, the reviewer suggests/claims that if we follow the updates
> > > $$
> > > w^+=w - \nabla^ 2 R_m(w_{m_{0}})^{-1} \nabla R_n(w),
> > > $$
> > > for the stage that we deal with $n$ samples, the iterates will converge to the optimal solution of that subproblem at a rate of $c^k$, where $c\in(0,1)$ and is independent of problem parameters. However, our analysis does not support this claim, and it is not clear to us under what conditions on $m_0$ this claim holds. To achieve such a fast linear convergence rate for the above algorithm that works for any value of $n$ and without requiring any adaptation step for the Hessian approximation, we need to develop a new analysis. We basically need to understand under what conditions on $m_0$ the method the reviewer has suggested converges at $1/2^k$ or a similar rate. Our current analysis studies the conditions required for the superlinear convergence of BFGS. By satisfying the same conditions, one cannot ensure that the algorithm proposed by the reviewer converges linearly.
> > >
> > > Note that we do not claim that achieving a linear rate of the form $c^k$ for the method proposed by the reviewer is not possible, but we simply believe that such result would definitely require a different condition on $m_0$, that could be possibly worse the one provided for our AdaQN method. This is not a conclusion that we can reach by the current analysis, as in this paper, we are not analyzing the algorithm suggested by the reviewer.
> > >
> > > In summary, we agree with the reviewer that one can exploit other approximations of Newton's method that locally converge at a rate of $1/2^k$ to solve the subproblems efficiently. However, in this paper, our focus is on studying the BFGS method and showing how it can achieve a similar sample complexity as the one for Ada Newton, without requiring $\log (N)$ Hessian inverse computations. We will highlight this point in the revised paper to clarify any possible ambiguity.
> > >
> > > Next, we answer the specific questions raised by the reviewer.
> > >
> > >  **Is it true that the rest of the analysis in the paper would still go through if the convergence guarantee in Proposition 2 had $(1/2)^{t_n}$ (linear) instead of $(1/t_n)^{t_n}$ (superlinear)?**
> > >
> > > Suppose a second-order method has a linear convergence rate, where its contraction factor is a constant and independent of problem parameters (such as the condition number and the dimension). In that case, it could solve each subproblem with a constant number of iterations.  But, as discussed above, we first need to find such a method, and then we need to identify the local conditions required to achieve such a fast linear convergence rate. Indeed, as these local conditions would be different from ours, the final condition on $m_0$ would be different too. In Propositions 3 and 4, we explicitly show the requirements for the initial point and initial Hessian approximation matrix to obtain the local superlinear convergence rate of $(1/t_n)^{t_n}$ for quasi-Newton methods. We use these local neighbor conditions to derive the lower bound on $m_0$. Hence, if we replace our analysis with some linearly convergent method (with rate $(1/2)^k$), we have to study under what conditions this fast local rate result holds, and using that result, we can identify the lower bound on $m_0$. Thus, the rest of the analysis in our paper needs to be modified if we replace the superlinear convergence rate with a linear convergence rate suggested by the reviewer.
> > >
> > > **Do you agree that this change makes the difference between "linear" and "superlinear" convergence?**
> > >
> > > Yes, that would be a linear rate, but as mentioned above, that local linear rate requires different local conditions.
> > >
> > > **Do you believe it is true that Algorithm 1 without any Hessian updates (lines 7 and 8) achieves the convergence guarantee $(1/c)^{t_n}$ for some absolute constant $c > 1$ (e.g., $c = 2$ ) under the assumptions of Proposition 2?**
> > >
> > > As we discussed above, the required conditions for which the updates
> > > $$
> > > w^+=w - \nabla^ 2 R_m(w_{m_{0}})^{-1} \nabla R_n(w)
> > > $$
> > > lead to a linear convergence rate of $(1/c)^{t_n}$ for some absolute constant $c > 1$ (e.g., $c = 2$ ) are not studied in our paper. We agree with the reviewer that it should be possible to analyze this algorithm, but our current analysis does not guarantee that with the conditions in Proposition 2, one can show that the above update (which uses the same Hessian inverse approximation $\nabla^ 2 R_m(w_{m_{0}})^{-1}$ for all the iterates and stages) would converge at a fast linear rate of $(1/c)^{t_n}$. Hence, the answer to your question is not positive, as we believe our analysis does not hold for your suggested method.
> > >
> > > We think that the necessary condition on $m_0$ for your suggested algorithm could be more strict than ours, as it uses one Hessian approximation for all iterations and all stages. Moreover, we believe that studying such an algorithm is beyond the scope of this paper, as in this work, we only aim to show the effectiveness of the BFGS method.

---

> > > > ### Comment · Reviewer_RQFB · 2021-08-24
> > > > **Response about superlinearity**
> > > >
> > > > Thank you to the authors for the response.
> > > >
> > > > As the authors are I'm sure aware, there is considerable literature on achieving linear convergence when using Newton's method with a subsampled Hessian approximation (e.g., [15] cited by the paper, but also [BBN19,RKM19] to name a few). In fact, even the paper [23], which the authors use for Proposition 2, can be used to establish such a convergence guarantee. See Section 4.2 and Appendix H (of the arxiv version). While those linear convergence results are seemingly established for the BFGS method, the algorithm in its first iteration is actually the same as a simple Newton's method with an approximate Hessian.
> > > >
> > > > For example, the assumptions in your Proposition 2 correspond to setting $\epsilon=1/50$, $\delta=1/7$ and $r=1/2$ in Theorem 5.4 of [23]. Following the analysis around equations (146)-(152) of [23], this implies a convergence rate of $1.1\cdot(1/4)^t$ if we keep the initial Hessian approximation: We simply use the equation (148) for k=0 and then note that if we do not update the Hessian approximation, then the initial conditions are still trivially preserved after one step. Then, we convert to convergence in function value to obtain the rate. Thus, we effectively skip all of the "superlinear" analysis from [23], and get a linear convergence rate under the same condition on $m_0$ as in Proposition 2. Remarkably, with this rate, only $t_n=2$ iterations (instead of $3$) are needed for the adaptive sample size analysis to go through. Note that this type of linear convergence analysis for subsampled Newton has appeared in many works, and is standard in the literature, I'm only using [23] to illustrate the point for the authors.
> > > >
> > > > This is why I find the claim "exploiting superlinear convergence" highly misleading, because it implies that you obtain something more that could not have been achieved with standard linear convergence results for subsampled Newton methods, but that in fact is not the case.
> > > >
> > > > [BBN19] Raghu Bollapragada, Richard H. Byrd, and Jorge Nocedal. "Exact and inexact subsampled Newton methods for optimization." IMA Journal of Numerical Analysis 39.2 (2019): 545-578.
> > > >
> > > > [RKM19] Roosta-Khorasani, Farbod, and Michael W. Mahoney. "Sub-sampled Newton methods." Mathematical Programming 174.1 (2019): 293-326.

---

> > > > > ### Author Response · Authors · 2021-08-24
> > > > > **Response to Reviewer RQFB**
> > > > >
> > > > > We thank the reviewer for the clarification.
> > > > >
> > > > > We never state anywhere in the paper that one can not achieve a similar complexity without exploiting the superlinear convergence rate. As we mentioned in our previous response, one can achieve a similar result by proposing an algorithm that achieves a linear rate of the forms $c^k$, if $c$ is independent of problem parameters. To derive our bounds, we used the superlinear rate of BFGS, but one can use another method with a linear rate to achieve a similar complexity. We never state in the paper that one must exploit the superlinear rate for achieving our complexity. We simply say if you use the superlinear rate analysis of BFGS, you will obtain what we have shown. We don't believe that this sentence is misleading by any means. Specifically, **we never state that only with exploiting superlinear rate one can achieve our results**. Even in the title of the paper, we state exploiting "local convergence of BFGS method".
> > > > >
> > > > > If the reviewer believes that this point should be highlighted in the paper, we would be more than happy to clarify this point in the revised paper by including a remark. But, it does not change the paper's main contribution, which is proposing a method that improves the complexity of Ada Newton. We don't believe there is any adaptive sample size method in the literature that provably and rigorously improves the complexity of the Ada Newton method. Indeed, one can exploit the linearly convergent methods suggested by the reviewer to introduce an adaptive sample size second-order method that outperforms Ada Newton, but AdaQN is the first method that achieves this goal.

---

### Decision · Program_Chairs · 2021-09-27

**Decision:**

Accept (Poster)

**Comment:**

This paper focuses on characterizing rates of convergence for Quasi-Newton method applied to empirical risk minimization. Traditional stochastic Quasi-Newton methods have not enjoyed guarantees that are faster than their first order counterparts. This paper uses an adaptive sampling scheme to achieve global superlinear convergence. All reviewers thought the idea of combining adaptive sampling with Quasi-Newton methods is interesting and appreciated the thorough global convergence analysis. The reviewers did however raise a few concerns including: (1) one reviewer had concerns about the claim that the proposed approach “exploits the superlinear convergence” of quasi-Newton methods, (2) lack of discussion of some related literature, (3) mismatch in m0 between theory and practice, and (4) a variety of other technical discussions. The authors provided a thorough response and the reviewers had a lively discussion with the authors and together. As a result some of the above issues were resolved and some reviewers raised their score. Issue number (1) however remained and that reviewer increased their score contingent on this issue being revised in the final version. I agree for the most part with the reviewers that the paper is interesting and clearly written with nice results. But also concur with them about the technical issues and that claims of exploiting superlinearity are a bit misleading. Therefore I recommend acceptance with the requirement that a clear discussion on superlinearity must be highlighted in the paper.